

**Hydraulic Shortcuts Increase the Connectivity of Arable Land Areas to**
**Surface Waters**
Urs Schönenberger[1] and Christian Stamm[1]
[1]Eawag, Swiss Federal Institute of Aquatic Science and Technology, 8600 Dübendorf, Switzerland.
**Abstract**
Surface runoff represents a major pathway for pesticide transport from agricultural areas to surface
waters. The influence of man-made structures (e.g. roads, hedges, ditches) on surface runoff
connectivity has been shown in various studies. In Switzerland, so-called hydraulic shortcuts (e.g.
inlets and maintenance manholes of road or field storm drainage systems) have been shown to
influence surface runoff connectivity and related pesticide transport. Their occurrence, and their
influence on surface runoff and pesticide connectivity have however not been studied systematically.
To address that deficit, we randomly selected 20 study areas (average size = 3.5 km$^2$) throughout the
Swiss plateau, representing arable cropping systems. We assessed shortcut occurrence in these study
areas using three mapping methods: field mapping, drainage plans, and high-resolution aerial images.
Surface runoff connectivity in the study areas was analysed using a 2x2 m digital elevation model and
a multiple-flow algorithm. Parameter uncertainty affecting this analysis was addressed by a Monte
Carlo simulation. With our approach, agricultural areas were divided into areas that are either directly
connected to surface waters, indirectly (i.e. via hydraulic shortcuts), or not connected at all. Finally,
the results of this connectivity analysis were scaled up to the national level using a regression model
based on topographic descriptors.



Inlets of the road storm drainage system were identified as the main shortcuts. On average, we found
0.84 inlets and a total of 2.0 manholes per hectare of agricultural land. In the study catchments
between 43 and 74 % of the agricultural area is connected to surface waters via hydraulic shortcuts.
On the national level, this fraction is similar (54 %).
These numbers suggest that transport through hydraulic shortcuts is an important pesticide flow path
in a landscape where many engineered structures exist to drain excess water from fields and roads.
However, this transport process is currently not considered in Swiss pesticide legislation and
authorisation. Therefore, current regulations may fall short to address the full extent of the pesticide
problem. Overall, the findings highlight the relevance of better understanding the connectivity
between fields and receiving waters and the underlying factors and physical structures in the
landscape.





## 1. Introduction

Agriculture has been shown to be a major source for pesticide contamination of surface waters (Stehle
and Schulz 2015; Loague, Corwin, and Ellsworth 1998). Pesticides are known to pose a risk to aquatic
organisms and to cause biodiversity losses in aquatic ecosystems (Malaj et al. 2014; Beketov et al.
2013). For implementing effective measures to protect surface waters from pesticide contamination,
the relevant transport processes have to be understood better.
Pesticides are lost to surface waters through various pathways from either point sources or diffuse
sources. In current research, surface runoff (Holvoet, Seuntjens, and Vanrolleghem 2007; Larsbo et al.
2016; Lefrancq et al. 2017), preferential flow through macropores into the tile drainage system
(Accinelli et al. 2002; Leu et al. 2004a; Reichenberger et al. 2007; Sandin et al. 2018), and spray drift
(Carlsen, Spliid, and Svensmark 2006; Schulz 2001; Vischetti et al. 2008) are considered of major
importance. Other diffuse pathways like leaching into groundwater and exfiltration into surface
waters, atmospheric deposition or aeolian deposition are usually less important.
Past research showed that different catchment parts can largely differ in their contribution to the
overall pollution of surface waters (Pionke et al. 1995; Leu et al. 2004b; Gomides Freitas et al. 2008).
This is the case for soil erosion or phosphorus, but also for pesticides. Areas largely contributing to the
overall pollution load are called critical source areas (CSAs). Models delineating such CSAs assume
that those areas fulfill three conditions (Doppler et al. 2012): i) They represent a substance source (e.g.
pesticides, soil, phosphorus), ii) they are hydrologically active (e.g. occurrence of surface runoff), and
iii) they are connected to a water body.
Linear landscape structures, such as hedges, ditches, tile drains, or roads have been shown to be
important features for the connectivity within a catchment (Fiener, Auerswald, and Van Oost 2011).
Undrained roads were reported to intercept flow paths, to concentrate and accelerate runoff, and
therefore also to influence pesticide connectivity within a catchment (Carluer and De Marsily 2004;
Dehotin et al. 2015; Heathwaite, Quinn, and Hewett 2005; Payraudeau et al. 2009). Additionally,
Lefrancq et al. (2013) showed that undrained roads act as interceptor of spray drift, possibly leading to



significant pesticide transport during subsequent rainfall events when intercepted pesticides are
washed off the roads.
However, such linear structures and the related connectivity effects exhibit substantial regional
differences due to natural conditions or various aspects of the farming systems. In contrast to other
countries, many roads in agricultural areas in Switzerland are drained by stormwater drainage systems
(Alder et al. 2015). Inlets of stormwater drainage systems are also found directly in fields (Doppler et
al. 2012; Prasuhn and Grünig 2001). Since those stormwater drainage systems were reported to
shortcut surface runoff to surface waters, those structures were called *hydraulic shortcuts* or short-
circuits. Doppler et al. (2012) showed in a small Swiss agricultural catchment that hydraulic shortcuts
were creating connectivity of remote areas to surface waters and had a strong influence on pesticide
transport. Only 4.4 % of the catchment area was connected directly to surface waters, while 23 % was
connected indirectly (i.e. via hydraulic shortcuts). For the same catchment, Ammann et al. (2020)
showed that the uncertainty of a pesticide transport model could be reduced by 30 % by including
catchment-specific knowledge about hydraulic shortcuts and tile drainages.
The occurrence of hydraulic shortcuts and their influence on catchment connectivity has only been
studied for a few other catchments in Switzerland. Prasuhn and Grünig (2001) found that only 3.2 %
of the arable land in five small catchments were connected directly to surface waters, while 62 % were
connected indirectly. Consequently, 90 % of the sediment lost to surface waters was transported
through shortcuts.
To our knowledge, these two studies are the only ones systematically assessing the occurrence of
hydraulic shortcuts and their influence on (sediment) connectivity. However, since these studies only
covered a small total area in specific regions, it remains unknown if these findings are generally valid
for Swiss agricultural areas.
Two other studies in Switzerland addressed connectivity on a larger scale using a modelling approach.
Both indicated that more areas were connected through shortcuts than directly. Bug and Mosimann
(2011) estimated 12.5 % of the arable land in the canton of Basel-Landschaft to be connected directly
to surface waters, and 35 % to be connected indirectly. Later, Alder et al. (2015) created a national



connectivity map of erosion risk areas. They estimated that 21 % of the agricultural area is connected
directly to surface waters and 34 % indirectly. In both studies, generalizing assumptions on the
occurrence of hydraulic shortcuts were made (e.g. classification of roads as drained by shortcuts or as
undrained, based on their size). Since only for small areas the occurrence of hydraulic shortcuts was
effectively known, these assumptions are quite uncertain as also stated by Alder et al. (2015).
In summary, previous studies on hydraulic shortcuts were either restricted to small study areas in a
specific region, or were based on generalizing assumptions, lacking a spatially explicit consideration
of hydraulic shortcuts. This study aims for a systematic, spatially distributed, and representative
assessment of hydraulic shortcut occurrence on Swiss agricultural areas. Based on this assessment we
aim on quantifying the influence of hydraulic shortcuts on surface runoff connectivity and pesticide
transport. We focused our study on arable land, since this is the largest type of agricultural land with
common pesticide application in Switzerland.
Our research questions therefore are:

1) How widespread do hydraulic shortcuts occur in Swiss arable land areas?

2) What is their relevance for surface runoff connectivity and for surface-runoff related pesticide

transport?

**Shortcut definition**
We define a hydraulic shortcut as *a man-made structure increasing and/or accelerating the process of*
*surface runoff reaching surface waters (i.e. rivers, streams, lakes) or making this process possible in*
*the first place*. In this study, we focused on the following structures (example photos can be found in
Figure S 1 to Figure S 12 in the SI):

111       A) Storm drainage inlets on roads, farm tracks and crop areas

B) Maintenance manholes of storm drainage systems or tile drainage system on roads, farm

tracks and crop areas

C) Channel drains and ditches on roads, farm tracks and crop areas



If one of these structures is present, we defined this as a *potential shortcut*. If surface runoff can enter
the structure and if the structure is drained to surface waters or to a wastewater treatment plant, this is
defined as a *real shortcut*. Other processes that are sometimes referred to as hydraulic shortcuts (e.g.
tile drains) are not considered in this study. Tile drains have already received considerable attention in
pesticide research and the transport to tile drains includes flow through natural soil structures.



## 2. Material and Methods

## 2.1.    Selection of study areas

We selected 20 study areas (Table 1) representing arable land in the Swiss plateau and the Jura
mountains (Fig. 1). This selection was performed randomly on a nationwide small-scale topographical
catchment dataset (BAFU 2012). The probability of selection was proportional to the total area of
arable land in the catchment as defined by the Swiss land use statistics (BFS 2014). Random selection
was performed using the pseudo-random number generator Mersenne Twister (Matsumoto and
Nishimura 1998).
On average, the study areas have a size of 3.5 km$^2$ and are covered by 59 % agricultural land. The
agricultural land mainly consists of arable land (74 %) and meadows (21 %). The mean slope on
agricultural land is 4.9 degrees and the mean annual precipitation amounts to 1159 mm yr$^{-1}$. A
comparison of important catchment properties of the study areas to the corresponding distribution of
all Swiss catchments with arable land demonstrated that the study areas represent the national
conditions well (see Figure S 14).
**Table 1: Catchment properties of the 20 study areas. Fractions of agricultural area and of arable land were**
**determined from BFS (2014). Mean slope of agricultural areas was determined from BFS (2014) and Swisstopo**
**(2018). Mean annual precipitation was determined from Kirchhofer and Sevruk (1992).**

| ID | Location | Canton | Receiving water | Area (km²) | Fraction of agricultural area | Fraction of arable land | Mean slope of agricultural areas in the catchment (deg) | Mean annual precipitation (mm/yr) |
|---|---|---|---|---|---|---|---|---|
| 1 | Böttstein | AG | Bruggbach | 3.3 | 52 % | 30 % | 8.5 | 1187 |
| 2 | Ueken | AG | Staffeleggbach | 2.0 | 42 % | 39 % | 7.6 | 1164 |
| 3 | Rüti b. R. | BE | Biberze | 2.2 | 29 % | 11 % | 11.2 | 1403 |
| 4 | Romont | FR | Glaney | 3.4 | 78 % | 48 % | 4.0 | 1344 |
| 5 | Meyrin | GE | Nant d'Avril | 10.0 | 49 % | 31 % | 3.2 | 1133 |
| 6 | Boncourt | JU | Saivu | 5.9 | 44 % | 23 % | 5.5 | 1093 |
| 7 | Courroux | JU | Canal de Bellevie | 2.8 | 82 % | 75 % | 2.9 | 1082 |
| 8 | Hochdorf | LU | Stägbach | 2.4 | 84 % | 59 % | 4.1 | 1213 |
| 9 | Müswangen | LU | Dorfbach | 3.0 | 79 % | 61 % | 4.0 | 1482 |
| 10 | Fleurier | NE | Buttes | 1.0 | 24 % | 11 % | 9.6 | 1538 |
| 11 | Lommiswil | SO | Bellacher Weiher | 3.8 | 50 % | 40 % | 6.8 | 1388 |
| 12 | Illighausen | TG | Tobelbach | 1.9 | 54 % | 30 % | 1.8 | 1122 |
| 13 | Oberneunforn | TG | Brüelbach | 3.3 | 69 % | 52 % | 4.2 | 968 |
| 14 | Clarmont | VD | Morges | 2.4 | 75 % | 70 % | 5.3 | 1163 |
| 15 | Molondin | VD | Flonzel | 4.2 | 74 % | 65 % | 5.9 | 1064 |



| | | | | | | | | |
|---|---|---|---|---|---|---|---|---|
| 16 | Suchy | VD | Ruiss. des Combes | 3.3 | 72 % | 63 % | 5.6 | 1026 |
| 17 | Vufflens | VD | Venoge | 2.8 | 39 % | 30 % | 5.7 | 1006 |
| 18 | Buchs | ZH | Furtbach | 3.9 | 57 % | 48 % | 4.9 | 1182 |
| 19 | Nürensdorf | ZH | Altbach | 2.3 | 59 % | 44 % | 3.6 | 1225 |
| 20 | Truttikon | ZH | Niederwisenbach | 5.1 | 66 % | 49 % | 4.6 | 960 |
| | **Mean** | | | **3.5** | **59 %** | **44 %** | **4.9** | **1159** |


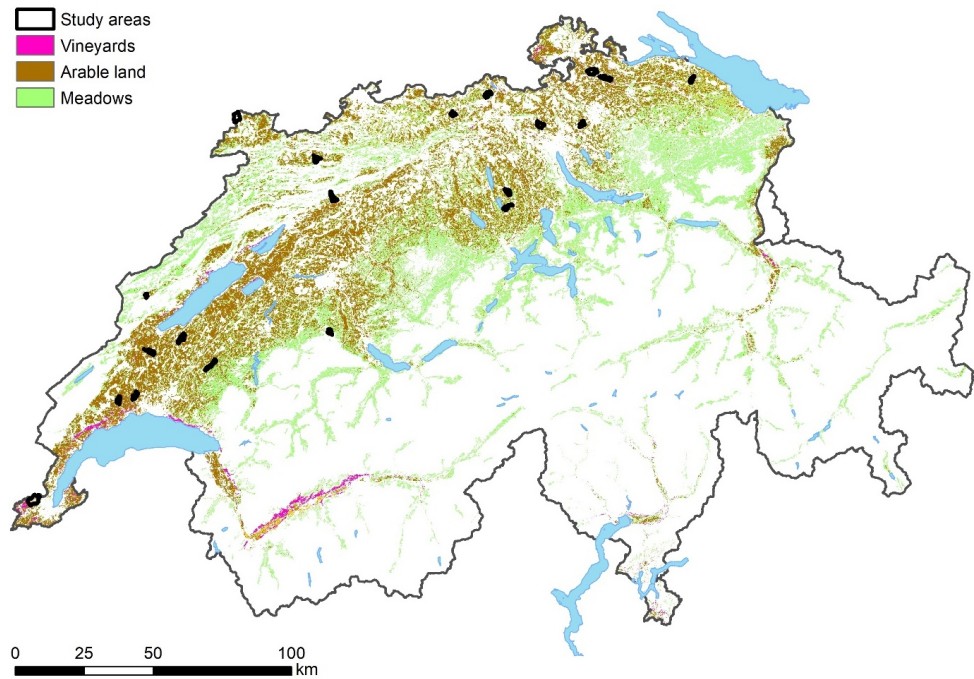


**Figure 1: Study areas (black) and distribution of arable land (brown), vineyards (pink), and meadows (green) across**
**Switzerland. Source: Swisstopo (2010); BFS (2014)**

## 2.2. Assessment of hydraulic shortcuts

**Shortcut location and type**
We mapped the location and types of potential shortcuts in each study area by combining three
different methods.
i) *Field survey*: Field surveys were performed between August 2017 and May 2018 (details see Table
S 4). In a subpart of each study area, we walked along roads and paths and mapped all the potential
shortcut structures. The starting point was selected randomly, and we mapped as much as we could

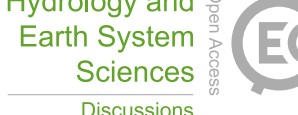

within one day. Consequently, the field survey data only cover a part of the catchment. For each of the
potential shortcuts we recorded its location, as well as a set of properties using a smartphone and the
app "Google My Maps". This included a specification of the type of the shortcut (e.g. inlet, inspection
chamber, ditches, channel drains), its lid type (e.g. grid, sealed lid, lid with small openings), and its lid
height relative to the ground surface. A list of all possible types can be found in the supporting
information (Table S 1 to Table S 3).
ii) *Drainage plans*: For all municipalities covering more than 5 % of a study area we asked the
responsible authorities to provide us with their plans of the road storm drainage systems and the
agricultural drainage systems. For 38 and 26 of the 46 municipalities concerned we received road
storm drainage system plans and tile drainage system plans, respectively. Reasons for missing data are
either that the responsible authorities did not respond or that data on the drainage systems were not
available. From the plans, we extracted the locations of shortcuts and, if available, the same properties
were specified as in the field survey.
iii) A*erial images*: Between August 2017 and August 2018 (details see Table S 4) we acquired aerial
images of the study areas with a ground resolution of 2.5 to 5 cm. We used a fixed-wing UAV (eBee,
Sensefly, Cheseaux-sur-Lausanne) in combination with a visible light camera (Sony DSC-WX220,
RGB). The study areas were fully covered by the UAV imagery, with the exception of larger
settlement areas, forests, and lakes, and of no-fly zones for drones (e.g. airports). The UAV images
were processed to one georeferenced aerial image per study area using the software Pix4Dmapper 4.2.
In the no-fly zones of the study areas Meyrin (Geneva), Buchs (Zürich), and Nürensdorf (Zürich) we
used aerial images provided by the cantons of Geneva (Etat de Genève 2016) and Zürich (Kanton
Zürich 2015). Ground resolutions were 5 cm, and 10 cm respectively. Using ArcGIS 10.7, we gridded
the aerial images, scanned by eye through each of the grid cells, and marked all potential shortcut
structures manually. If observable from the aerial image, the same properties as for the field survey
were specified for each potential shortcut structure.
We combined the three datasets originating from the three methods to a single dataset. If a potential
shortcut structure was only found by one of the mapping methods, its location and type were used for
the combined dataset. If it was found by more than one of the mapping methods, we used the location



and type of the mapping method that we expected to be the most accurate. For the location
information, this is UAV imagery, before field survey, and maps. For the type specification, this is
field survey, before UAV imagery, and maps.
**Assigning shortcuts to different landscape elements**
In order to understand better where hydraulic shortcuts occur the most, we assigned them to different
landscape elements. Using the topographic landscape model of Switzerland "swissTLM3D"
(Swisstopo 2010) we defined five landscape elements: Paved roads, unpaved roads, fields, settlements,
and other areas (e.g. railways, other traffic areas, forests, water bodies, wetlands, single buildings). For
all landscape elements except roads and railways, shortcuts were assigned to their landscape elements
by a simple intersection. However, shortcuts belonging to road or railway drainage systems are in
many cases not placed on the road or railway directly, but on the adjacent agricultural land or
settlement. Therefore, shortcuts were assigned to the landscape elements road or railway if they were
within a 5 m buffer.
In addition, we correlated the density of shortcuts per study area to different study area properties. We
selected study area properties that we expected to have explanatory power: density (length per area) of
paved roads, density of unpaved roads, density of surface rivers, density of subsurface rivers, mean
annual precipitation, and mean slope on agricultural areas.
**Drainage of shortcuts**
A potential shortcut only turns into a real one if it is drained to surface waters by pipes or other
connecting structures, such as ditches. Therefore, using the plans provided by the municipalities, we
investigated where potential shortcuts drain to. They were allocated to one of the following categories
of recipient areas: surface waters, wastewater treatment plants/combined sewer overflow, infiltration
areas (e.g. forest, infiltration ponds, fields, grassland), or unknown.







## 2.3.    Surface runoff connectivity model


We created a surface runoff connectivity model to estimate which fraction of potentially pesticide-
loaded surface runoff originating on agricultural land is reaching surface waters via hydraulic shortcuts
in comparison to direct transport. The model is based on the concept of critical source areas (CSAs).
An area is defined as a CSA if 1) pesticides are applied on the area, 2) it is connected to surface
waters, and 3) it is hydrologically active (i.e., generating fast flow processes transporting pesticides to
streams). This model mainly focuses on the first two of these elements, while the question whether an
area is hydrologically active is only addressed partially because many relevant information such as soil
properties are not available at the national scale.
The model (see Figure 2) distinguishes *source areas* on which surface runoff is produced, and
*recipient areas* on which surface runoff ends up. A *connectivity model* connects those areas by routing
surface runoff through the landscape. These model parts are conceptually described in more detail in
the section "model structure". In the section "model parametrization", we describe how we
parametrized the model and how we assessed the uncertainty of model output given the parameter
uncertainty. In the last section "hydrological activity", we explain the testing for systematic
differences in the hydrological activity between areas with direct or indirect connectivity.
**Model structure**

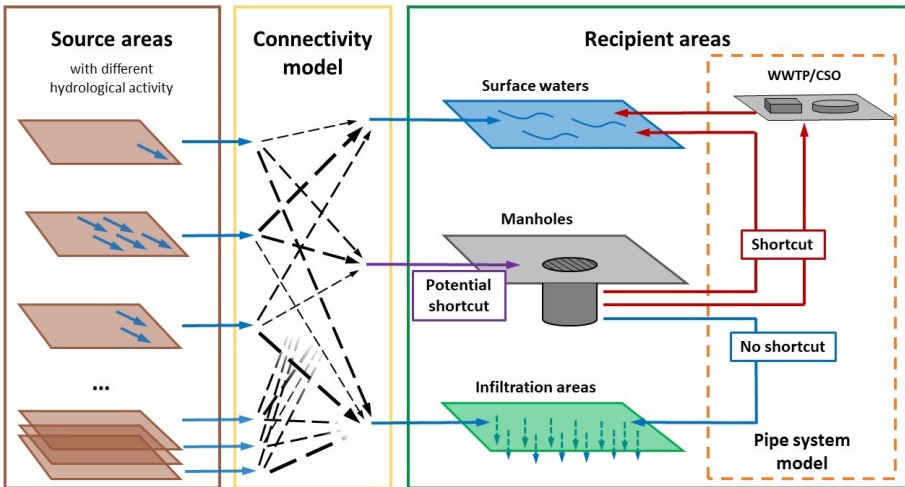


**Figure 2: Structure of the surface runoff connectivity model**


***Source areas.*** All crop areas on which pesticides are applied should in theory be considered as source
areas. However, a highly resolved spatial dataset of land in a crop rotation for our study areas is
lacking. Therefore, we considered the total extent of agricultural areas (i.e. arable land, meadows,
vineyards, orchards, and gardening) as source areas, since those areas could be derived in high
resolution. The extent of agricultural areas was defined by subtracting all non-agricultural areas
(forests, water bodies, urban areas, traffic areas, and other non-agricultural areas) as defined by the
national topographical landscape model SwissTLM3D (Swisstopo 2010) from the total area of each
study area. According to the Swiss proof of ecological performance (PEP), pesticide usage within a
distance of 6 m from a river, and within 3 m from hedges and forests is prohibited. The extent of
agricultural areas was reduced accordingly except along forests (parameters *river spray buffer*, *hedge*
*spray buffer*).
***Recipient areas.*** Surface runoff generated on a source area and routed through the landscape can end
up in three different types of landscape elements, referred to as recipient areas: Surface waters,
infiltration areas (i.e. forests, hedges, internal sinks), and shortcuts. The extent of surface waters
(rivers that have their course above the surface, lakes, and wetlands), was defined by the
SwissTLM3D model as was the extent of forests and hedges. Since forests and hedges are known to
infiltrate surface runoff (Sweeney and Newbold 2014; Schultz et al. 2004; Bunzel, Liess, and
Kattwinkel 2014; Dosskey, Eisenhauer, and Helmers 2005) we assumed that forests with a certain
width (parameter *infiltration width*) act as an infiltration area. Hedges were assumed either to act as
infiltrations areas, or to have no effect on surface runoff. Accordingly, the parameter *hedge*
*infiltration*, was varied between yes (hedges act as infiltration areas) and no (hedges don't act as an
infiltration areas).
Internal sinks in the landscape were defined using the 2x2m digital elevation model (Swisstopo 2018).
All sinks larger than two raster cells and deeper than a certain depth (parameter *sink depth*) were
defined as internal sinks. All other sinks were filled completely.
Shortcuts were defined in two different ways (parameter *shortcut definition*): In definition A, all inlets,
ditches, and channel drains were considered as potential shortcuts. In definition B, manholes lying in
internal sinks were additionally considered as potential shortcuts. Potential shortcuts were defined to



act as real shortcuts if they are known to discharge to surface waters or wastewater treatment plants.
From the drainage plans of the municipalities, we know that most of the inlets discharge into either a
surface water body or a wastewater treatment plant. Therefore, also potential shortcuts with unknown
drainage location were assumed to act as real shortcuts. Potential shortcuts discharging into forests or
infiltration structures were assumed not to act as shortcuts and were not used in the model. Shortcut
recipient areas were defined as the raster cells of the digital elevation model on which the shortcut is
located and all the cells directly surrounding it (see Figure S 13 in the SI).
***Connectivity model***. For modelling connectivity we used the TauDEM model (Tarboton 1997) which
is based on a D-infinity flow direction approach. As an input we used a 2x2m digital elevation model
(DEM) (Swisstopo 2018). This DEM was modified as follows: We assumed that only those internal
sinks that were defined as sink recipient areas (see above) effectively act as sinks. Therefore, firstly,
all sinks were filled, and sink recipient areas were carved 10 m into the DEM. Secondly, all other
recipient areas (shortcuts, forests, hedges, surface waters) were carved between 10 and 50 m into the
DEM. Carving the recipient areas into the DEM ensured that surface runoff reaching a recipient area
was not routed further on to another recipient area. Thirdly, to account for the effect of roads
accumulating surface runoff (Heathwaite, Quinn, and Hewett 2005), roads were carved into the DEM
by a given depth defined by the parameter *road carving depth*.
The modified DEM, the source areas, and the recipient areas were used as an input into the TauDEM
tool "D-Infinity upslope dependence". Like this, each raster cell belonging to a source area was
assigned with a probability to be drained into one of the three types of recipient areas.
The connectivity of a source area may depend on the flow distance to surface waters. For longer flow
distances, water has a higher probability to infiltrate before it reaches a surface water. Therefore, for
each source area raster cell, we calculated the flow distance to its recipient area using the tool "D-
infinity distance down". Source areas with flow distances longer than the parameter *maximal flow*
*distance* were then defined as not connected.
**Model parametrization and sensitivity analyses**



The model parameters mentioned in the section above vary in space and time. Since this variability
could not be addressed with the selection of a single parameter value, we performed a Monte Carlo
simulation with 100 realizations. The probability distributions of the parameters are provided in Table
2. The bounds or categories of these distributions were based on our prior knowledge about the
hydrological processes involved, about structural aspects (e.g. depths of sinks), and on our experience
from field mapping. The parameters *river spray buffer* and *hedge spray buffer* were assumed constant
according to the guidelines of the Swiss proof of ecological performance (PEP). For the parameter
*maximal flow distance,* all possible flow distances were evaluated.
To assess the influence of single parameters on our modelling results, we performed a local sensitivity
analysis against a benchmark model (one realization of the model with a specific parameter set, see
Table 2). When selecting the benchmark model parameter set, we kept the changes in the digital
elevation model small (i.e. *road carving depth* = 0 cm, *sink depth* = 10 cm), and the maximal flow
distance was not reduced (*maximal flow distance* = ∞). For the other model parameters, we selected
the values that we assumed to be the most probable in reality. For the local sensitivity analysis, each of
the model parameters was varied individually within the same boundaries as for the Monte Carlo
analysis.
**Table 2: Summary of parameter distributions used for the Monte Carlo analysis and parameter values used as a**
**benchmark for the sensitivity analysis. PEP: Swiss proof of ecological performance.**

| Parameter | Handling of parameter uncertainty | Distribution | Bounds / Categories | Benchmark model |
|---|---|---|---|---|
| Sink depth | Monte Carlo & sensitivity analysis | Uniform distribution | 5 cm ≤ x ≤ 100 cm | 10 cm |
| Infiltration width | Monte Carlo & sensitivity analysis | Uniform distribution | 6 m ≤ x ≤ 100 m | 20 m |
| Road carving depth | Monte Carlo & sensitivity analysis | Uniform distribution | 0 cm ≤ x ≤ 100 cm | 0 cm |
| Shortcut definition | Monte Carlo & sensitivity analysis | Bernoulli distribution | [Definition A; Definition B] | Definition A |
| Hedge infiltration | Monte Carlo & sensitivity analysis | Bernoulli distribution | [yes; no] | Yes |
| River spray buffer | Assumed as certain, based PEP guidelines | Constant | 6 m | 6 m |
| Hedge spray buffer | Assumed as certain, based PEP guidelines | Constant | 3 m | 3 m |
| Maximal flow distance | Calculation of all possible flow distances | - | 2m ≤ x ≤ ∞ | ∞ |





**Hydrological activity**
As mentioned earlier, a critical source area has to be hydrologically active, i.e. surface runoff has to be
generated on that area. Runoff generation depends on many variables (e.g. crop types, soil types, soil
moisture, rain intensity) for which no data are available in most of our study areas and which are
strongly variable over time. Since we are interested in the general relevance of shortcuts, we focused
on the question whether there is a systematic difference in the hydrological activity between areas
directly or indirectly connected to streams.
For soil moisture, we tested for such differences by calculating the distribution of the topographic
wetness index (TWI) for the source areas of the benchmark model. We calculated the TWI as follows,
using the "Topographic Wetness Index" tool of the TauDEM model (Tarboton 1997):
$$\mathrm{TWI} = \frac{\ln(a)}{\tan(\beta)}$$

The local upslope area $a$, and the local slope $\beta$ were calculated using the D-infinity flow direction
algorithm that was already used for the surface runoff connectivity model. As an input, we used the
source areas and the modified DEM as specified for the surface runoff connectivity model.
The formation of surface runoff on agricultural areas is also influenced by their slope. Therefore, we
calculated the distribution of slopes for source areas draining to different destinations. For this we used
the slopes from the Swiss digital elevation model (Swisstopo 2018).
For other variables (e.g. crop type, rain intensity), there is no indication for such systematic
differences. Therefore, we assumed that they do not differ systematically between areas draining to
different recipient areas.





## 2.4.    Extrapolation to the national level
**Extrapolation of the local connectivity model**
In order to assess the relevance of shortcuts for the whole country, we developed a model for
extrapolating the results from our study areas (local surface runoff connectivity model, LSCM) to the
national scale.
*Selection of explanatory variables:* We calculated a list of catchment statistics based on nationally
available geodatasets that could serve as explanatory variables. As catchment boundaries, the polygons
from the national catchment dataset (BAFU 2012) were used. Catchment statistics included fraction of
forests, fraction of agricultural area, road density (total, paved, unpaved), water body density (total,
rivers, lakeshores), mean annual precipitation, mean slope of agricultural areas, and area fractions
(direct, indirect, not connected) as reported by the national erosion connectivity model (NECM) (Alder
et al. 2015). Details on the datasets used for calculating those catchment statistics can be found in
Table S 5 of the supporting information.
We created a linear regression between each of those catchment statistics to the fractions of
agricultural areas directly, indirectly, and not connected to surface waters, as reported by the LSCM
($f_{LSCM,dir}$, $f_{LSCM,indir}$, $f_{LSCM,nc}$). The strongest correlations were found for the fractions of agricultural areas
directly, indirectly, and not connected to surface waters, as reported by the NECM ($f_{NECM,dir}$, $f_{NECM,indir}$,
$f_{NECM,nc}$, see Table S 8). Therefore, we used them as explanatory variables for building an extrapolation
model of our local results to the national scale.
The model predictions for each catchment have to fulfil specific boundary conditions: Firstly, the sum
of areal fractions of the three types of recipient areas k per catchment c has to equal one ($\sum_{k=1}^{K} f_{k,c} =$
1), and secondly, area fractions cannot be negative ($f_{k,c} \geq 0$). To ensure these conditions, we
performed the model fit after a unit simplex data transformation. The resulting modelling approach is
shown in Figure 3. Mathematical details are provided in the SI (chapter S1.5).





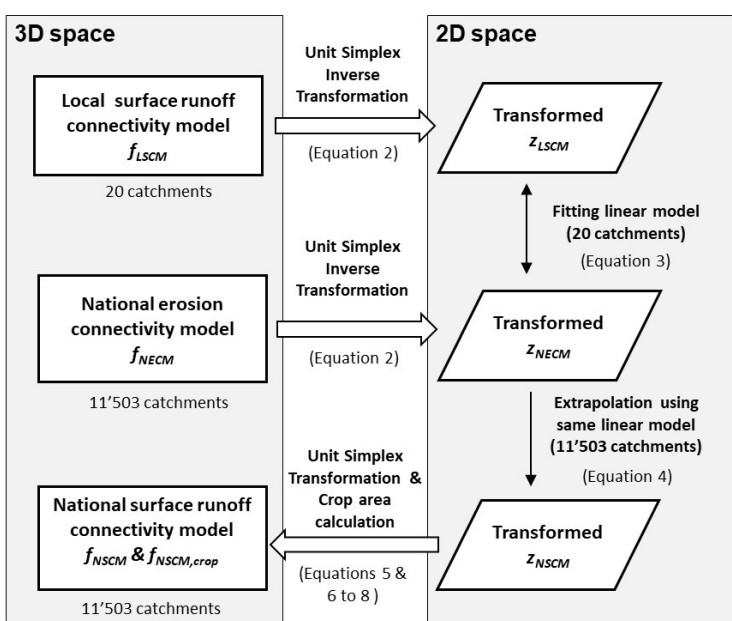



Figure 3: Extrapolation of the local surface runoff connectivity model (LSCM) to the national scale (NSCM) using a unit simplex transformation approach.

As a result, we obtained a national surface runoff connectivity model (NSCM). The NSCM provides
an estimate for the fractions of agricultural areas directly, indirectly, and not connected to surface
waters ($f_{NSCM,dir}$, $f_{NSCM,indir}$, $f_{NSCM,nc}$) for the catchments of the national catchment dataset. Since in the
NECM mountainous regions of higher altitudes are excluded, those areas are also excluded in the
NSCM.
**Connectivity of crop areas**
Since there are no high-resolution datasets of crop areas yet available in Switzerland, we considered
the total extent of agricultural areas for building the local surface runoff connectivity model and
extrapolation to the national scale. These areas include areas with rare pesticide application, such as
meadows, which are not expected to act as source areas (except in special cases such as fighting weeds
such as bitter dock (*Rumex obtusifolius L.*)).
The Swiss land use statistics dataset (BFS 2014) is a raster dataset with a resolution of 100 m, dividing
agricultural areas into different categories (e.g. arable land, vineyards, meadows). On the national



scale, the usage of such a lower-resolution dataset is more reasonable. Hence, we used this dataset for
calculating fractions of connected crop areas.
The fractions of directly, indirectly, and not connected crop areas per total agricultural area per
catchment c ($f_{NSCM,crop,c}$) were calculated as follows:
$$f_{NSCM,crop,c} = f_{NSCM,c} \cdot r_{crop,c} \tag{6}$$
With $r_{crop}$ being the ratio of crop area to total agricultural area in a catchment:
$$r_{crop,c} = \frac{A_{crop,c}}{A_{crop,c} + A_{mead,c}} \tag{7}$$
$$A_{crop,c} = A_{arab,c} + A_{vin,c} + A_{orch,c} + A_{gard,c} \tag{8}$$
with:        $A_{crop,c}$ = Crop area in catchment c (ha)

$A_{mead,c}$ = Meadow and pasture areas in catchment c (ha)

$A_{arab,c}$ = Arable land area in catchment c (ha)

$A_{vin,c}$ = Vineyard area in catchment c (ha)

$A_{orch,c}$ = Orchard area in catchment c (ha)

$A_{gard,c}$ = Gardening area in catchment c (ha)





## 3. Results

## 3.1.  Occurrence of hydraulic shortcuts

In the following section, we first show the results of the field mapping campaign for manholes (inlets,

maintenance manholes) followed by the results for channel drains and ditches. Afterwards we present

results on the accuracy of our mapping methods.

**Manholes**

In total, we found 8213 manholes, corresponding to an average manhole density of 2.0 ha$^{-1}$ (min.:

0.51 ha$^{-1}$, max.: 4.4 ha$^{-1}$; Table 3). Forty-two percent of the manholes mapped were inlets. A plot

showing the density of manholes mapped per catchment and manhole type can be found in Figure S 15

in the supporting information.

For roughly half of the inlets and maintenance manholes we were able to identify a drainage location.

Both manholes types discharge in almost all cases into surface waters, either directly (87 % of inlets,

63 % of maintenance manholes) or via wastewater treatment plants or combined sewer overflow (12 %

of inlets, 37 % of maintenance manholes). Only 1.4 % of the inlets and no maintenance manhole at all,

were found to drain to an infiltration area, such as forests or fields.

**Table 3: Number of manholes found on agricultural areas of the study areas per shortcut category and drainage location.**

| Drainage location | Inlets | | Maintenance manholes | | Other manholes | | Unknown type | |
|---|---|---|---|---|---|---|---|---|
| | Count | Fraction | Count | Fraction | Count | Fraction | Count | Fraction |
| Surface waters | 1568 | 46 % | 1205 | 29 % | 0 | 0 % | 0 | 0 % |
| WWTP/CSO | 218 | 6 % | 705 | 17 % | 0 | 0 % | 0 | 0 % |
| Infiltration areas | 26 | 1 % | 0 | 0 % | 0 | 0 % | 0 | 0 % |
| Unknown | 1615 | 47 % | 2227 | 54 % | 31 | 100 % | 618 | 100 % |
| **Total** | **3427** | **100 %** | **4137** | **100 %** | **31** | **100 %** | **618** | **100 %** |

Most of the inlets mapped (90 %) are located on paved or unpaved roads (min: 66 %, max: 100 %;

Table 4). Only very few inlets (2.8 %) are found directly on fields. In contrast, maintenance manholes

are found much more often on fields (mean: 21 %, min: 0 %, max: 42 %) and therefore less often on

paved or unpaved roads (mean: 52 %, min: 39 %, max: 88 %). The fractions of inlets and maintenance




manholes belonging to a certain landscape element for each study area can be found in Figure S 18 in
the supporting information.
**Table 4: Percentage of manholes found on a certain type of landscape element. The category "other areas" integrates**
**several types of landscape elements: railways, other traffic areas, forests, water bodies, wetlands, and single buildings.**

|  | Paved roads | Unpaved roads | Settle-ments | Fields | Other areas |
|---|---|---|---|---|---|
| **Inlets** | 79 % | 10 % | 5.5 % | 2.8 % | 2.2 % |
| **Maintenance manholes** | 52 % | 7.2 % | 16 % | 21 % | 4.5 % |


We correlated the densities of inlets and maintenance manholes per study area with possible
explanatory variables. Only the density of paved roads was significantly correlated to the density of
inlets ($R^2 = 0.33$, $p = 0.008$) and maintenance manholes ($R^2 = 0.37$, $p = 0.005$). Details can be found in
Table S 6 and Table S 7.
**Channel drains and ditches**
In addition to manholes, we also mapped channel drains and ditches. With the exception of the study
areas Meyrin (4.2 m ha$^{-1}$) and Buchs (4.0 m ha$^{-1}$) these structures were rarely found (< 1.2 m ha$^{-1}$; see
Figure S 16). In Meyrin and Buchs, most channel drains and ditches (98 % of the total length) drain to
surface waters, and only few of them to infiltration areas (2 %).
**Mapping accuracy**
The results above were generated using three different mapping methods (*field survey*, *UAV images*,
and *drainage plans*). These methods differ in their ability to identify and classify a potential shortcut
structure correctly and in the study area they cover. We determined the accuracy of the mapping
methods aerial images and drainage plans using the field survey method as a ground truth (see Table
5) for those parts of the study areas where all three methods were applied. Since channel drains and
ditches were rare, this assessment was only performed for manholes.
The recall (i.e. the probability that a potential shortcut is found by a mapping method) was limited for
the aerial images method (53 % for inlets, and 62 % for maintenance manholes), and even lower for
the drainage plans method (32 % for inlets, and 21 % for maintenance manholes). However, identified





shortcuts were in most of the cases classified correctly (accuracy: 93 % to 94 % for aerial images,
88 % to 89 % for drainage plans).
For the entire study areas, Figure 4 shows the number of potential shortcuts identified by the three
mapping methods. Despite a low recall, aerial images identified the largest number of potential
shortcuts. This is due to the large spatial coverage by the aerial images method. Since the overlap
between the three methods is small (only 32 % of the inlets and 15 % of the maintenance manholes
were found by more than one method), each of the methods was important to determine the total
number of potential shortcuts in the study areas. Because the aerial images and drainage plans have a
low recall, but cover large parts of the study areas that were not assessed by the field survey, the
numbers reported above are a lower boundary estimate.
**Table 5: Recall and classification accuracies of the mapping methods aerial images and drainage plans. The recall**
**corresponds to the probability that a potential shortcut is found by the mapping method. Percentages indicate the**
**recall of each individual mapping method. In brackets, the recall of the combination of both methods is given. The**
**accuracy corresponds to the sum of true positive fraction and true negative fraction.**

| Mapping method | Manhole type | Recall | True positives | False positives | True negatives | False negatives | Accuracy |
|---|---|---|---|---|---|---|---|
| Aerial images | Inlets | 53 % (60 %) | 61 % | 1.3 % | 33 % | 4.9 % | 94 % |
| | Maintenance manholes | 62 % (69 %) | 32 % | 5.3 % | 61 % | 1.3 % | 93 % |
| Drainage plans | Inlets | 32 % (60 %) | 67 % | 4.5 % | 22 % | 6.6 % | 89 % |
| | Maintenance manholes | 21 % (69 %) | 20 % | 7.1 % | 68 % | 5.3 % | 88 % |


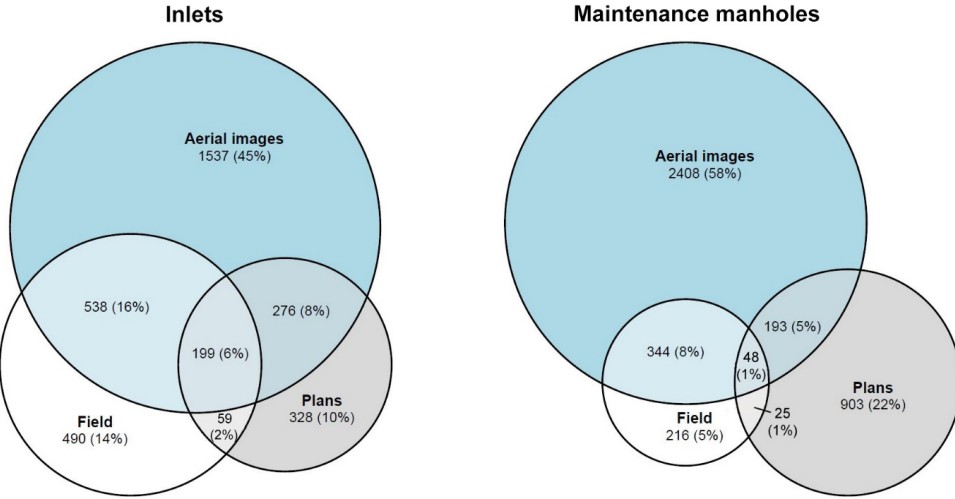


**Figure 4: Number of inlets (left) and maintenance manholes (right) identified by the different mapping methods.**



## 3.2. Surface runoff connectivity

### 3.2.1. Study areas

Based on the Monte Carlo analysis of the surface runoff connectivity model, we estimated the

fractions of agricultural areas that are connected directly, indirectly, or not at all to surface waters. To

illustrate the variability resulting from these Monte Carlo (MC) runs, Figure 5 shows the output of

three MC simulations (MC28, MC41, and MC40) for Molondin. These simulations correspond to the

5 %, 50 %, and 95 % quantile of the median fraction of indirectly connected per total connected

agricultural area over all study catchments. While certain areas change their classification depending

on the model parametrisation (e.g. letters A to C), for other parts of the catchment, the results of the

MC simulations are very consistent (e.g. letters D to F). Overall, the results show that not only

agricultural areas close to surface waters (e.g. letter D) are connected to surface waters. Hydraulic

shortcuts also create surface runoff connectivity for areas far away from surface waters (e.g. letter E).

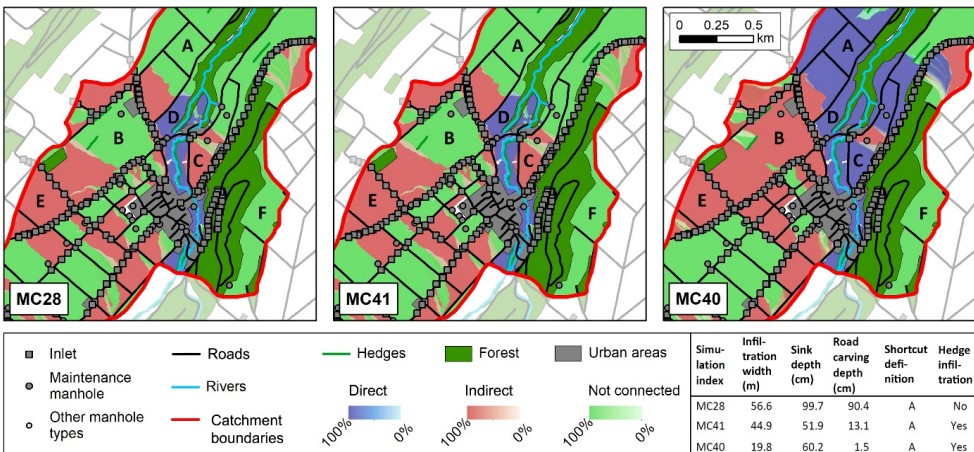

**Figure 5: Directly connected (blue), indirectly connected (red) and not connected (green) areas resulting from three example Monte Carlo (MC) simulations for a part of the study area Molondin. The simulations represent approximately the 5 % (MC28), 50 % (MC41), and 95 % (MC40) quantiles with respect to the resulting median fractions of indirectly connected per total connected area over all study catchments. The parameters of the example MC simulations are shown on the bottom right. Source of background map: Swisstopo (2010)**

In order to assess the importance of hydraulic shortcuts, we calculated the fraction of indirectly

connected area to the total connected area. Across all Monte Carlo simulations, the median of this

fraction over all study catchments ranges between 43 % and 74 % (mean: 57 %, median: 58 %; Figure





5). Despite considerable uncertainty, the results demonstrate that a large fraction of the surface runoff
connectivity to surface waters is established by hydraulic shortcuts.

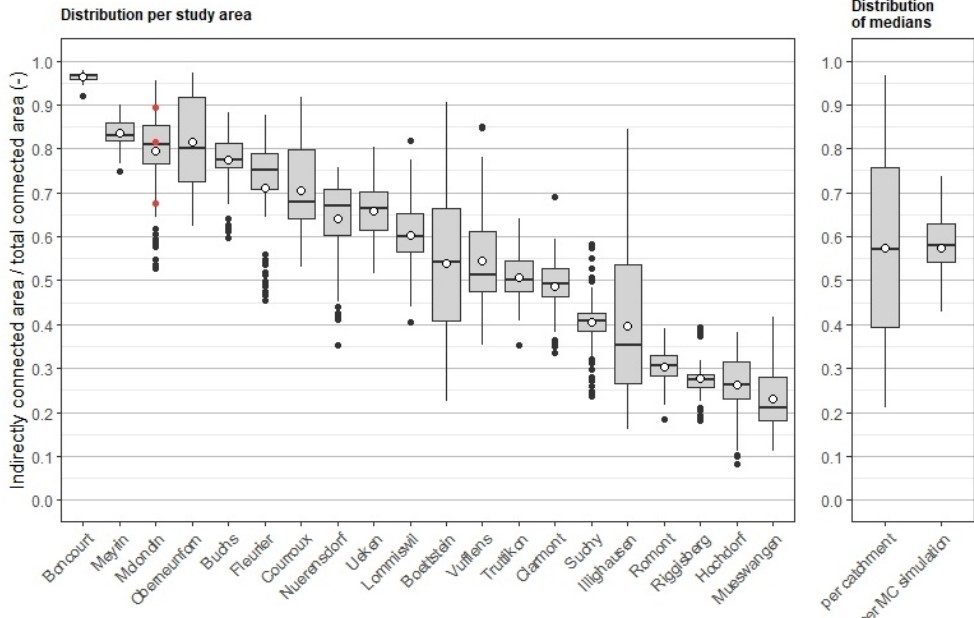


**Figure 6: Left: Fractions of indirectly connected areas per total connected areas as calculated by the Monte Carlo**
**analysis for each study area. White dots indicate the means of the distributions. The red dots indicate the results of the**
**example Monte Carlo simulations (MC28, MC41, and MC 40) shown in Figure 5. Right: Distribution of medians of**
**fractions of indirectly connected areas per total connected areas per study catchment and per Monte Carlo**
**simulation.**
However, this fraction varies strongly between the study areas, ranging from 21 % in Müswangen to
97 % in Boncourt. Although the occurrence of hydraulic shortcuts is a prerequisite of indirect
connectivity, high manholes densities are not necessarily leading to high fractions of indirect
connectivity in a catchment. The densities of inlets and maintenance manholes show only a weak
positive correlation to the catchment medians of the fraction of indirectly connected areas (inlets: $R^2$ =
0.11, p = 0.15; maintenance manholes: $R^2$ = 0.08, p = 0.23; see Table S 8). By contrast, the two study
areas with high channel drain and ditch densities (Meyrin and Buchs) show high fractions of indirect
connectivity. Similarly, the density of surface waters is strongly negatively correlated to the fraction of
indirect connectivity ($R^2$ = 0.51, p < 0.001).  This suggests that line elements like channel drains,
ditches and surface waters usually have an influence on connectivity if they occur in a catchment. By
contrast, the influence of point elements seems to depend a lot on the surrounding landscape structure.



As a further consequence of the structural differences between the study areas, not all of them reacted
the same way to changes in model parameters of the Monte Carlo analysis. For example, the fraction
of indirectly to total connected areas in the study area Boncourt was quite insensitive to changes in
model parameters. Since Boncourt has a very low water body density, only small areas are connected
directly, independent of the model parametrization. The study area Illighausen, on the other hand,
reacted very sensitively (range of results = 68 %). Since Illighausen is a very flat catchment, changes
in the sink depth parameter had a large influence on the estimated fractions of direct and indirect
connectivity.
So far, we only reported on the fraction of indirectly connected per total connected area. In Table 6,
we additionally report the fractions of total agricultural area connected directly, indirectly, and not at
all to surface waters. On average, we estimate between 5.5 % and 38 % (mean: 28 %) of the
agricultural area to be connected directly, 13 % to 51 % (mean: 35 %) to be connected indirectly, and
12 % to 77 % (mean: 37 %) not to be connected to surface waters. However, the variation between the
catchments is much larger than the variation of the Monte Carlo analysis.
**Table 6: Fractions of directly, indirectly, and not connected agricultural areas in our study catchments. The first row**
**represent the mean fraction over all catchments and Monte Carlo simulations. The second row represents the median**
**of the median over all catchments per MC simulation. The third row represents the median of the median over all MC**
**analyses per catchment. In brackets, the minimum and the maximum median are given.**

| Statistic | Fraction of directly connected agricultural area $f_{dir}$ | Fraction of indirectly connected agricultural area $f_{indir}$ | Fraction of not connected agricultural area $f_{nc}$ | Fraction of indirectly per total connected area $f_{fracindir}$ |
|---|---|---|---|---|
| Mean | 28 % | 35 % | 37 % | 57 % |
| Median per MC simulation | 25 % (5.5 %; 38 %) | 38 % (13 %; 51 %) | 32 % (12 %; 77 %) | 58 % (43 %; 74 %) |
| Median per catchment | 26 % (1.8 %; 70 %) | 37 % (12 %; 60 %) | 35 % (3.9 %; 53 %) | 57 % (21 %; 97 %) |


**Sensitivity analysis**
In the previous section, variation due to model parameter uncertainty was addressed globally by
analysing the variation of Monte Carlo simulation results. To analyse which model parameters have
the largest influence on our model results, we tested the local model parameter sensitivity on our
benchmark model. Our results show that the fraction of indirectly to total connected area reacts most
sensitive to changes in the road carving depth parameter. The difference between the minimal and
maximal fraction reported was 17 %. Results were also sensitive to the parameters shortcut definition





(14 %) and sink depth (13 %). Infiltration width (4.3 %) and hedge infiltration (2.5 %) had only a
minor influence on the fraction reported. Detailed results can be found in Figure S 22 and Figure S 23
in the supporting information. We also analysed how the fraction of indirect to total connected areas
changed with flow distance. However, the sensitivity was rather small (details see Figure S 24).
**Hydrological activity**
Systematic differences in hydrological activity between directly and indirectly connected areas would
have a major influence on the interpretation of our connectivity analysis. We therefore tested for such
differences by calculating the distributions of slope and topographic wetness index on these areas.
The distributions of both, slope and topographic wetness index were very similar for directly,
indirectly, and not connected areas (see Figure S 25 and Figure S 26). Only the slope of not connected
areas was found to be slightly smaller than the slope of connected areas. Hence, we could not identify
any systematic differences in the factors affecting hydrological activity between directly and indirectly
connected areas.

### 3.2.2.  Extrapolation to the national level
We created a model for extrapolating the results of our study areas to the national level, using area
fractions of the national erosion connectivity model (NECM) (Alder et al. 2015) aggregated to the
catchment scale as explanatory variables. The area fractions of the NECM were transformed such that
they fit the area fractions of the local surface runoff connectivity model (LSCM) resulting from the
Monte Carlo analysis in our study areas. The resulting dataset is called the national surface runoff
connectivity model (NSCM). As depicted in Figure 7, the differences in the mean and standard
deviation of directly connected and not connected area fractions were strongly reduced by this
transformation in our study areas. Differences in mean and standard deviation of indirectly connected
area fractions were already small before the transformation and did not change substantially.



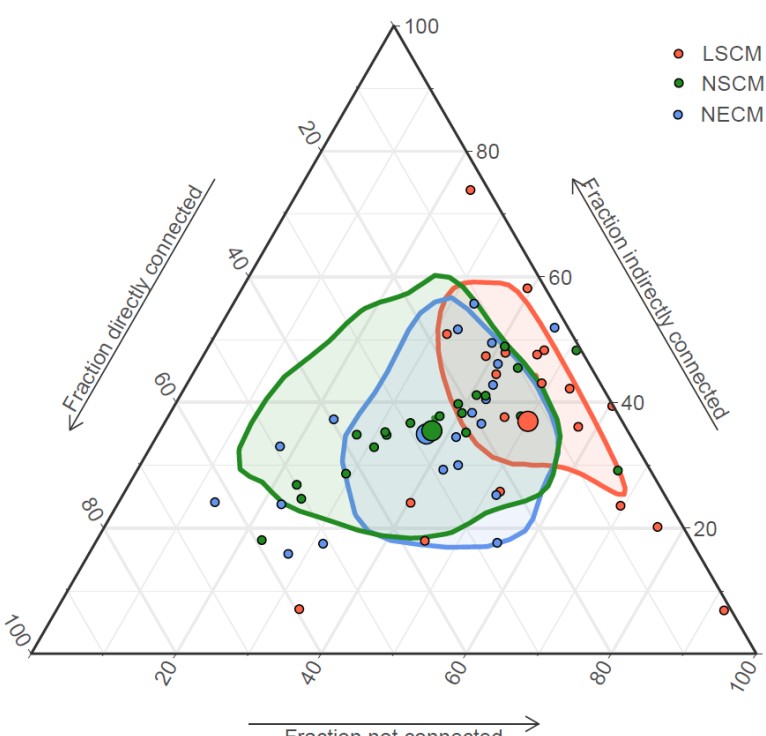


**Figure 7: Fractions of directly connected ($f_{dir}$), indirectly connected ($f_{indir}$), and not connected areas ($f_{nc}$) per total agricultural area for the local surface runoff connectivity model (LSCM, blue), national erosion connectivity model (NECM, red), and national surface runoff connectivity model (NSCM, green) in the 20 study areas. Small blue circles represent the catchment medians of all Monte Carlo simulations of the LSCM, small red circles represent the data reported by the NECM, and small green circles represent the catchment medians of the NSCM. Large circles represent the means of the LSCM (blue), NECM (red), and NSCM data (green). Shaded areas represent normal Kernel density estimates of the LSCM, NECM, and NSCM data.**

Using the transformation derived from our study areas, we extrapolated the results of the local surface
runoff connectivity model to the national scale, resulting in a national surface runoff connectivity
model (NSCM) aggregated to the catchment scale. It covers all catchments of the valley zones, hill
zones and lower elevation mountain zones. Using land use data, we additionally calculated the fraction
of agricultural crop area per total agricultural area of each catchment. Multiplication of this fraction
with the NSCM resulted in an estimate of connected crop areas on the national scale. Half of the Swiss
agricultural areas in the model region are crop areas (i.e. arable land, vineyards, orchards, horticulture)
and therefore potential pesticide source areas (details see Figure S 27). Twenty six percent of crop
areas (13 % of total agricultural area) are connected directly, 34 % (17 % of total agricultural area)
indirectly, and 40 % (20 % of total agricultural area) not at all. From the total connected crop area,
54 % are connected indirectly. These results are similar to those obtained for the 20 study areas (see


above). Mean fractions of directly and indirectly connected areas are a bit smaller in the national scale
estimation than for the 20 study areas (-2.0 %, and -1.9 %), while the fraction of not connected area is
a bit larger (+3 %). The fraction of indirectly connected crop area per total connected crop area is
slightly smaller (-2.6 %).
Compared to the national erosion connectivity model (NECM), the national surface runoff
connectivity model (NSCM) shows lower fractions of not connected crop areas (-7.2 %), but higher
fractions of directly connected crop areas (+6.2 %). The fractions of indirectly connected areas are
approximately the same between the two models (+1 %). Consequently, the fraction of indirectly
connected per total connected crop area is lower in the NSCM (-11 %).
Fractions of indirectly connected crop area per total agricultural area for all Swiss catchments in the
valley zones, hill zones and lower elevation mountain zones are shown in Figure 8. This map
corresponds to a risk map of pesticide transport via hydraulic shortcuts from agricultural areas to
surface waters. Areas of high risk for indirect pesticide transport are mainly found in the valley and
hill zones of the Swiss midlands, as well as in the Rhone valley. In higher zones (low mountain
zones), agricultural areas mainly consist of grassland (see Figure S 28). Therefore, higher zones pose a
low risk for pesticide transport, although their fraction of indirectly connected agricultural area can be
very high in certain regions, such as the Jura region (see Figure S 30 in the supporting information).
However, these regions still pose a risk for indirect transport of other pollutants, such as eroded soil or
nitrate to surface waters.



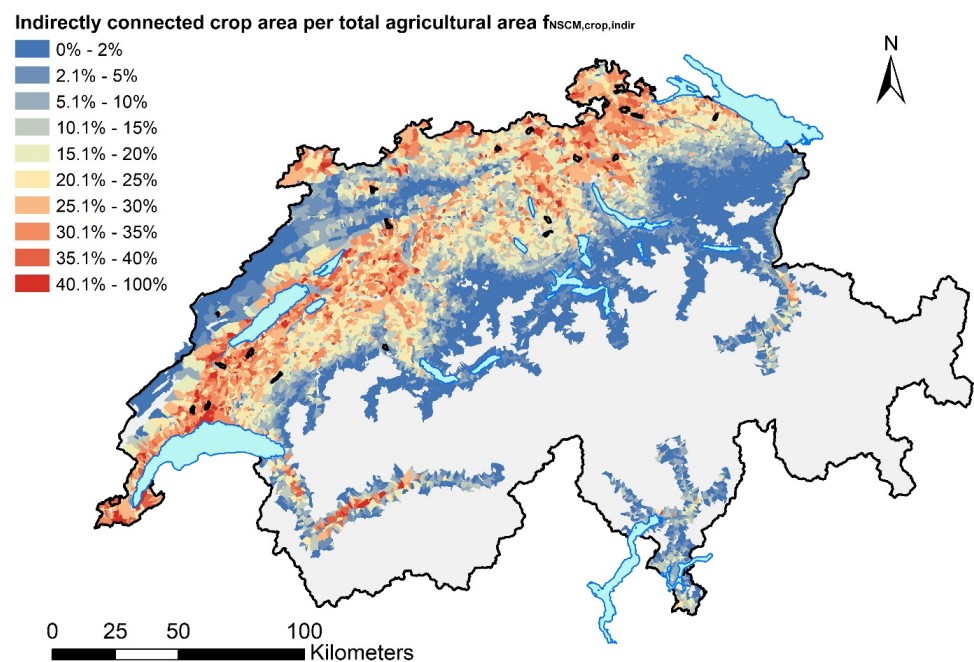


**Figure 8: Fraction of indirectly connected crop area per total agricultural area f_{NSCM,crop,indir} for all Swiss catchments**
**in the valley zones, hill zones and lower elevation mountain zones. Study areas are marked with black lines. Grey**
**areas represent higher elevation mountain zones that were excluded from the analysis. Source of background map:**
**Swisstopo (2010)**


## 4. Discussion
**Occurrence of hydraulic shortcuts**
Our study shows that storm drainage inlets and maintenance manholes are common structures found in
Swiss agricultural areas. While in neighbouring countries roads are often drained by ditches, Swiss
roads are usually drained by storm drainage inlets (Alder et al. 2015). It is therefore not surprising that
most of the inlets found in the study areas are located on roads. These findings are in accordance with
the only other study in Switzerland reporting numbers on storm drainage inlets (Prasuhn and Grünig

2001).

The vast majority of mapped storm drainage inlets were found to discharge to surface waters directly
or via wastewater treatment plants (WWTPs). Thus, the occurrence of an inlet is in most cases directly



related to a risk for pesticide transport to surface waters. The following three processes generate this
risk: Firstly, pesticide loaded surface runoff produced on crop areas can enter the inlet. Secondly,
spray drift deposited on roads can be washed off and enter the inlet. Thirdly, inlets can be oversprayed
during pesticide application, which is mainly considered probable for inlets located in the fields.
Although maintenance manholes were also found to discharge to surface waters directly or via
WWTPs, their occurrence does not directly translate into a risk for pesticide transport to surface
waters. In contrast to storm drainage inlets, maintenance manholes are not designed to collect surface
runoff. Their lids are usually closed or only have a small opening, significantly decreasing the risk of
surface runoff entering the manhole or of overspraying. In addition, lids of maintenance manholes in
fields are often elevated compared to the soil surface. Maintenance manholes on roads are (in contrast
to inlets) usually positioned such that concentrated surface runoff is bypassing them. However, as also
shown by Doppler et al. (2012), maintenance manholes can collect surface runoff from fields if they
are located in a sink or a thalweg and water is ponding above them during rain events. During our field
mapping campaign, we additionally found several damaged maintenance manholes that could easily
act as a shortcut.
Channel drains and ditches discharging into surface waters were rare in most study areas with two
exceptions. In Meyrin, the large length of these structures can be explained by the existence of a large
vineyard. Additionally, the density of manholes in this vineyard was higher than on the surrounding
arable land. This indicates that vineyards could generally have higher shortcut densities than arable
land. In Buchs, around 60 % of the channel drain and ditch length in the catchment are ditches at the
boundary between a ditches and a small streams. They are not appearing in the national topographic
landscape model (Swisstopo 2010) that was used for the definition of rivers and streams and did not
appear to be streams during field mapping or when analysing aerial images.
The number of mapped shortcuts represents a lower boundary estimate of the shortcuts present (see
results) and therefore leads to an underestimation of indirect connectivity. Probabilities for missing
shortcuts during our mapping campaign depend on their location. While aerial images were at almost
full coverage of the study areas, field mapping was performed mainly along roads. Drainage plans





were available more often along roads than on fields. Therefore, we expect that detection probability
of shortcuts is generally higher along roads than on fields. Besides coverage, various other factors
influence the detection probabilities of the mapping methods. Field mapping and aerial image
detection performance is reduced if shortcuts are covered. Along roads, this is mainly caused by
leaves, soil, and for aerial images also by trees and vehicles. On the fields, this is mainly caused by
soil or by crops. Detection performance of the aerial images method is additionally influenced by
image quality and ground resolution. Image quality is mainly influenced by wind and light conditions
during the UAV flights. In order to ensure high image quality, we planned UAV flights such that
weather conditions were favourable (low wind, slightly overcast). However, differences in image
quality between the study areas could not be completely avoided. Higher ground resolution could
further improve the data produced. Although detection performance is not expected to be limited by
the ground resolution used, higher resolution could improve the correct classification of shortcut types.
**Surface runoff connectivity**
Our study shows that around half of the surface runoff connectivity in our study areas, but also on the
national scale, is generated by hydraulic shortcuts. Surface runoff is considered one of the most
important processes for pesticide transport to surface waters. Consequently, a large amount of the
pesticide loads found in surface waters during rain events is expected to be transported by hydraulic
shortcuts. These findings are in accordance to the results of other studies investigating the influence of
hydraulic shortcuts on surface runoff connectivity (Alder et al. 2015; Prasuhn and Grünig 2001; Bug
and Mosimann 2011) and on pesticide transport (Doppler et al. 2012).
The fraction of indirect connectivity was found to be very different between study areas. The
variability introduced by the different properties of the study areas was larger than the variability
introduced by the different model parameters of the Monte Carlo analysis, indicating that our results
are robust against changes of our model parameters. Our model was most sensitive to changes of the
parameters *road carving depth*, *shortcut definition*, and *sink depth*. These parameters are discussed in
the following.





The parameter *road carving depth* accounts for the property of roads of collecting and concentrating
surface runoff. This effect is strongly dependent on microtopography, extremely variable in space, and
can therefore not be properly accounted for by a space-independent parameter. Usage of a higher
resoluted digital elevation model could however reduce the uncertainty on the effect of roads on
connectivity. Higher resolved digital elevation models would also help in capturing the influence of
other microtopographical features better. For example, small ditches or small elevations on the ground
can easily channel surface runoff. This can either direct surface runoff into a shortcut from areas not
modelled to drain to a shortcut, or vice versa. In Switzerland, a new digital elevation model with a
raster resolution of 0.5 m (swisstopo 2019) recently became available and could be used for this
purpose. This elevation model was not used within this study, since the study already had progressed
further by the time the dataset was published.
The model parameters *shortcut definition* (i.e. are maintenance manholes in a sink considered as a
shortcut) and *sink depth* are both related to the fate of surface runoff ponding in a sink. This indicates
that maintenance manholes in sinks could have an important influence on surface runoff connectivity
of agricultural areas. During our field mapping campaign, only few maintenance manholes in sinks
were investigated. It is therefore unclear if most maintenance manholes in sinks are capturing ponding
surface runoff, if surface runoff is usually infiltrating into the soil, or if it continues to flow on the
surface. Sensitivity of our model to the parameter *sink depth* additionally highlights that sinks can play
an important role for connectivity. Therefore, they should not be filled completely during GIS
analyses, as this is done by default by some flow routing algorithms.
Surface runoff is usually assumed to drain to the receiving water of its topographical catchment.
However, in various cases, the pipes draining hydraulic shortcuts were found to cross topographical
catchment boundaries. Consequently, surface runoff and related pesticide loads are transported to a
different receiving water than expected by the topographical catchment. This may be important to
consider when interpreting pesticide monitoring data from small catchments. Similar effects were
already reported for karstic aquifers or the storm drainage systems of urban areas (Jankowfsky et al.
2013; Luo et al. 2016).





**Hydrological activity**

We did not find any indication on systematic differences between the factors controlling hydrological

activities of directly and indirectly connected agricultural areas by analysing slope and topographic

wetness index. Those variables are a proxy for surface runoff formation, soil moisture, groundwater

level, but also physical properties of the soil (Sorensen, Zinko, and Seibert 2006; Ayele et al. 2020).

However, the hydrological activity of an agricultural area also depends on other factors that were not

quantitatively analysed, such as *rainfall intensities, crop types, soil management practices*, or the

presence of *tile drainage systems*.

*Rainfall intensities:* Because of the small size of the study areas and the close proximity between

directly and indirectly connected areas, systematic differences in rainfall intensities can be excluded.

*Crop types and soil management* can have a strong impact on runoff formation. These practices are

chosen by the farmers and there could be systematic differences of these variables. For example,

farmers aware of the effect of surface runoff and erosion on the pollution of surface waters might use

different cultivation methods or crops (e.g. conservation tillage) on fields close to surface waters than

on fields far away. This would lead to a higher probability of surface runoff formation on indirectly

connected areas compared to directly connected areas. However, different cultivation methods require

different farm machinery. Therefore, cultivation methods are often constrained by the machinery

available and farmers use the same cultivation method per crop for all of their fields. Consequently,

systematic differences in crop types or soil management between directly and indirectly connected

areas are unlikely. Nevertheless, in Switzerland, a national plot-specific crop type geodataset is

currently being developed. In the future, this dataset could give further insight into this question.

*Tile drainage systems:* Maintenance manholes and inlets found in the field often belong to a tile

drainage system. Therefore, fields on which maintenance manholes or inlets are located, have a higher

probability to be drained by tile drainage systems than other fields. This could lead to higher

infiltration capacities and consequently to reduced surface runoff on indirectly connected areas

compared to directly connected areas. However, since most of the inlets and manholes are located



along roads (see results) such differences would only have a minor effect on the overall surface runoff
connectivity.
**Extrapolation to the national level**
For extrapolating the results of our study areas to the national level, we used the national erosion
connectivity model (NECM) (Alder et al. 2015) since this dataset correlated best with the results of the
local connectivity model (LSCM). Alder et al. (2015) pointed out that the largest uncertainty of the
NECM is the classification of roads as drained or undrained, which was based on generalising
assumptions. The national surface runoff connectivity model (NSCM) combines the advantages of the
LSCM (consideration of field data on effective shortcut locations) and the NECM (modelling
shortcuts on the national scale). In addition, the NSCM also includes statistical information on crops
grown per catchment, which is not the case for the NECM. The result is an improved estimation of
surface runoff connectivity for crop areas on the national scale.
For creating the NSCM, all crop areas on which pesticides are commonly applied (arable land,
vineyards, orchards, horticulture) were assumed to contribute by the same amount to the pesticide
transport via surface runoff. However, these crop types are known to differ in the amounts of pesticide
applied (De Baan, Spycher, and Daniel 2015), in the amounts of surface runoff produced, and also
with respect to their connectivity to surface waters. This assumption could therefore be refined by
considering pesticide application data and by investigating surface runoff connectivity in vineyards,
orchards and horticulture in more detail.
In contrast to the NECM, which reports connectivity on a 2x2 m raster, the NSCM is aggregated to the
catchment scale. Therefore, it cannot be used as an instrument for pinpointing critical source areas
within in a catchment, as this is the case for the NECM. However, the NSCM can indicate the risk
posed to the receiving waters of all Swiss catchments by direct or indirect surface runoff from crop
areas. Authorities could therefore use the NSCM to select high-risk catchments and prioritize
measures. Additionally, our results on the occurrence of hydraulic shortcuts could be used to improve
the current version of the NECM.



**Relevance in a broader geographical context**

This study focussed on the relevance of hydraulic shortcuts in Switzerland. To our knowledge, no
studies have systematically analysed the occurrence of hydraulic shortcuts in other countries.
Nevertheless, the available literature suggests that in some regions such man-made structures like
roads, pipes, or ditches may be important for connecting fields with the stream network (Lefrancq et
al. 2013; Gassmann, Lange, and Schuetz 2012; Bug and Mosimann 2011). Based on our findings, we
hypothesise that shortcuts are mainly important in areas with small field sizes. This increases the
density of linear structures such as roads for access.

**Implications for practice**

In Swiss plant protection[1] legislation and authorisation, the effect of hydraulic shortcuts on pesticide
transport is currently not considered. Pesticide application is prohibited within a buffer of 3 m along
open water bodies and according to the Swiss proof of ecological performance (PEP) vegetated buffer
strip have to at least 6 m wide. In contrast, along roads, a buffer of only 0.5 m is required. Hence, the
current Swiss legislation is protecting surface waters against direct, but not against indirect transport.
This contrasts with the results of this study, showing that approximately half of the surface runoff
related pesticide transport is occurring indirectly. This gap between legislation on direct and indirect
transport was already pointed out by Alder et al. (2015) for soil erosion.
The most evident measure based on the current legislation are vegetated buffer strips along drained
roads and around hydraulic shortcuts, infiltrating surface runoff before it reaches a shortcut. Generally,
measures increasing infiltration capacity on the field would reduce pesticide transport. Other measures
could aim on the shortcut structures themselves (e.g. construction of shortcuts as small infiltration
basins, drainage of shortcuts to infiltration basins, removal of shortcuts).

---

[1] In this study, we have been using the general term "pesticides" instead of "plant protection products" to make the text more readable. Since we only looked at substances used for plant protection in an agricultural context, the term "plant protection products" would have been more precise. The term "pesticides", however, also includes "biocides" which are substances for control of plants or animals used in a non-agricultural context and were not subject of this study. The substances addressed in this study are regulated in the Swiss plant protection legislation and authorisation.





Finally, pesticide transport via hydraulic shortcuts should be incorporated into the registration
procedure and be considered for the mandatory mitigation measures that go with a registration.
Models used in this context are currently only considering transport via direct surface runoff, erosion,
tile drainages, and spray drift (De Baan 2020).
**Further research**
Our results suggest that the presence of hydraulic shortcuts as well as the fraction of indirectly
connected areas are higher in vineyards than on arable land. Since this study focused mainly on the
latter, the sample size was too small for a quantitative analysis of vineyards. The fact that Swiss
vineyards usually have high road densities points into the same direction. In Swiss vineyards,
pesticides are applied more often and in larger amounts than on arable land (De Baan, Spycher, and
Daniel 2015). Therefore, an assessment of hydraulic shortcut relevance in vineyards is needed.
Hydraulic shortcuts are not only collecting surface runoff from target areas, but also from non-target
areas such as roads. As shown by Lefrancq et al. (2013), large amounts of spray drift can be deposited
on roads. In Switzerland, these deposits are expected to be washed off during rain events and to be
transported to surface waters via hydraulic shortcuts. Further research should aim on quantifying the
amounts of spray drift deposited on roads and transported to surface waters via hydraulic shortcuts.
Although model estimations can give insight of pesticide transport via hydraulic shortcuts on a large
scale, they have not been validated in the field. Targeted measurements on pesticide transport through
shortcuts are needed to provide evidence on the quantitative relevance of this flow path.





**5. Conclusions**
Our study shows that hydraulic shortcuts are common structures found in Swiss arable land areas of
the Swiss plateau. Shortcuts are found mainly along roads, but also directly in the field. The
connectivity analyses suggests that on average, around half of the surface runoff connectivity and
related pesticide transport to surface waters from arable land is caused by hydraulic shortcuts.
However, in Swiss pesticide legislation and pesticide authorisation, hydraulic shortcuts are currently
not considered. Therefore, current regulations may fall short to address the full extent of the problem.
The national surface runoff connectivity model developed in this study identifies high-risk catchments
for pesticide transport to surface waters via hydraulic shortcuts.
Overall, the findings highlight the relevance of better understanding the connectivity between fields
and the receiving water and the underlying factors and physical structures in the landscape. Further
research should aim on analysing the effect of hydraulic shortcuts on surface runoff on other types of
agricultural crops, such as orchards or vineyards. In addition, the current type of landscape analysis
should be complemented by measurements on actual pesticide concentrations and loads in hydraulic
shortcuts in the field.





## 6. Code availability

If the manuscript is accepted, the following code will be made available via https://opendata.eawag.ch/

(FAIR repository):

- • Code for random selection of study areas
- • Code for definition of agricultural areas

## 7. Data availability

If the manuscript is accepted, the following datasets will be made available via
https://opendata.eawag.ch/ (FAIR repository):

- • Study areas (geodataset)
- • Aerial images
- • Shortcut locations (geodataset)
- • Estimated fractions of directly and indirectly connected areas for all catchments in valley
- zones, hill zones and lower elevation mountain zones (results of the NSCM model)

## 8. Team list

Urs Schönenberger, Christian Stamm

## 9. CRedit author contribution statement

**Urs Schönenberger:** Conceptualization, Methodology, Investigation, Formal analysis, Software, Data
curation, Writing - original draft, Visualization
**Christian Stamm:** Conceptualization, Methodology, Writing - review & editing, Funding acquisition

## 10. Competing interests

Author Christian Stamm is a member of the editorial board of the HESS journal.



**11.   Acknowledgements**
The authors would like to thank Michael Döring, Diego Tonolla, and Matthew Moy de Vitry for the
help regarding UAV operation. We would like to thank Volker Prasuhn for the feedback provided to
our research approach, Andreas Scheidegger for his help regarding statistical modelling, and Max
Maurer for reviewing this manuscript. Furthermore, we would like to thank all municipalities, cantons,
cooperative associations, and engineering offices that provided drainage plans for this study. Finally,
we want to thank the federal office of the environment for the funding of this work (contract
00.0445.PZ I P293-1032).



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
