# Peer review of "Hydraulic Shortcuts Increase the Connectivity of Arable Land Areas to"

_Hydrology and Earth System Sciences, 2020_

## Referee Comment (RC1) · Anonymous Referee #1 · 1 Nov 2020

The manuscript presents a simple and interesting model application that demonstrates the importance of hydraulic short-cuts in connecting runoff from agricultural land to water courses in Switzerland. I found the manuscript well-written, and the results and figures nicely displayed. The level of documentation and attention to detail, particularly in the supplementary material, are impressive. The field data are even more impressive.

Overall I believe the manuscript can be divided into two parts. In the first section, the authors map the location of every potential drainage shortcut in 20 small Swiss catchments. This information is then used to model the direct and indirect runoff connectivity

of agricultural plots to water courses. In the second part, these results are extrapolated to the whole of Switzerland.

While I have only minor questions and comments about the first part of the manuscript, I have some criticism about second. As I explain in detail in the comments below, I do not agree with the methods used for extrapolating the catchment-based connectivity model to the national scale. In my opinion, the data from the 20 catchments already substantiates the point the authors are trying to make. In the end, I had the impression that the up-scaling only makes the paper lose traction and does not seem to be scientifically interesting.

In any case, I am looking forward to hear what the authors have to say about this issue and eventually to see this published.

Specific comments & technical corrections

L42: Please consider rephrasing to: "relevant process have to be understood".

L116-117: I am curious to why shortcuts that drain into surface waters or treatment plants are treated same by the model. If you are looking at pollutant transport, shouldn't there be a difference?

L182: Please consider rephrasing to: "In order to better understand..."

Figure 2: Please define WWTP/CSO in the legend.

L236: What are internal sinks?

L258: Regarding the connectivity model...Maybe I missed something, but how do you go from the upslope dependence output raster to defining if a cell is directly or indirectly connected?

L265: How was this "carving" performed? Did you use a stream burning algorithm?

L274-275 & Table 2: The 2m limit for the maximal flow distance seems unrealistically

low. Is there a reason why you chose this value?

L303: I think a reference to Beven and Kirkby (1979) should be given where you explain the TWI, not Tarboton 1997.

L329: What model output data did you use for the regression? The median of the MC simulations per catchment? Results from all simulations?

Here I must say that I found using the NECM as an explanatory variable somewhat strange. If I understand this right, you are fitting a linear model to predict the outcomes of your model (LSCM), based on the results of another model (NECM). But if the latter is such a good predictor of runoff connectivity, couldn't you just recommend using it at national scale? At least until you have enough data to parameterize the LSCM for all of Switzerland?

L355: Is meadow the correct term here?

L436-437: How?

L443: Here I had the impression you are changing the language with which you describe negative ("While certain areas change their classification. . .") or positive ("for other parts results are very consistent)" results. Please consider rephrasing.

Anyway, I found these results quite interesting. Would it be a good idea to look at where the model is consistent and where it is not? I mean, considering model uncertainty, which fields are consistently identified as highly connected? Could these areas be regarded at higher risk of pollutant transport than others? Moreover, if you find out where the models are inconsistent, you can try to figure out why?

Figure 5: I didn't understand the colour-ramp bars in the figure legend. Are they necessary?

L547-551: If you propagate the uncertainty in your linear model (e.g. by simulating posteriors of the slope and the intercept and then bootstrapping model predictions), it

is likely that these differences will be within your error bands. I guess my question here is: is your extrapolated national model sufficiently different from the NECM to justify its usefulness?

L652: I am curious: which kind of sink filling algorithm would you recommend in this case?

L693-697: Here you explain the improvements of the NSCM over the NECM regarding the representation of runoff connectivity, which was helpful. While I agree that the information on crop statistics might help your model, I do not see how the national map incorporates the advantages of the LSCM (i.e. all the impressive field data you collected). In the end I have the impression that this upscaling doesn't do justice to all the work you went through in the small catchments. Moreover, while you appropriately represent the uncertainty of the LSCM, this is somewhat neglected in the extrapolation to the national scale. Would you not expect grater errors in the NSCM than in the LSCM?

---

## Author Comment (AC1) · 12 Nov 2020

**Answer to Anonymous Referee #1**

We would like to thank Referee #1 for the constructive and valuable feedback. We mostly agree with the referee's comments and will address them individually in the following. We still think that the manuscript part about the up scaling to the national level is important and scientifically interesting and will explain our arguments in more detail below. Nevertheless, we agree on the criticism brought up concerning to this part and proposed some adaptations to address it.

**L42: Please consider rephrasing to: "relevant process have to be understood".**

We agree on this. Will be rephrased accordingly.

**L116-117: I am curious to why shortcuts that drain into surface waters or treatment plants are treated same by the model. If you are looking at pollutant transport, shouldn't there be a difference?**

We agree that there is a difference between these two processes. In waste water treatment plants pesticides are removed to a certain degree. However, during heavy rain events (i.e. the point in time when the largest pesticide loads are expected in shortcuts) rain water is often not reaching WWTPs but directed to surface waters through CSO. Pesticides have been reported to transport pesticides (Mutzner et al., 2020) and we expect the transport via CSO to be very similar to "normal" shortcuts. In addition, from our field studies we know transport via WWTPs/CSO is less important. Only 12 % of the inlets mapped drain to WWTPs/CSO while 87 % drain to surface waters (see L382-384). We therefore do not expect these differences to have a major influence on our results and decided to neglect them.

**L182: Please consider rephrasing to: "In order to better understand: : :"**

We agree on this. Will be rephrased accordingly.

**Figure 2: Please define WWTP/CSO in the legend.**

We will add a definition for these abbreviations in the legend.

**L236: What are internal sinks?**

Internal sinks are depressions from where water cannot flow on the soil surface to a receiving water, but infiltrates water locally. This term is for example also used in Frey et al. (2009).

**L258: Regarding the connectivity model: Maybe I missed something, but how do you go from the upslope dependence output raster to defining if a cell is directly or indirectly connected?**

For every Monte Carlo parameter combination, we ran the tool "D-Infinity upslope dependence" three times. Each time we used a different type of recipient area (surface waters, shortcuts, infiltration areas) as an input. This resulted in three upslope dependence output rasters – one each for each type of recipient areas. These rasters then defined if a cell is directly, indirectly, or not connected.

**L265: How was this "carving" performed? Did you use a stream burning algorithm?**

We carved the recipient areas (as defined in L243-257) into the DEM by rasterizing topographic data and lowering the corresponding cells. For the recipient area type "surface waters" this corresponds to a (very simple) stream burning algorithm. We did not impose an additional gradient towards the stream as it is done by some stream burning algorithms. We think that using such a gradient would not change our results, since the river course of the raw DEM and the burnt-in river course align very well.

**L274-275 & Table 2: The 2m limit for the maximal flow distance seems unrealistically low. Is there a reason why you chose this value?**

Yes, we agree that this is unrealistically low. However, for the maximal flow distance we simply calculated all possible values. Since our DEM has a resolution of 2m, this was simply the shortest flow distance that could be calculated. For analysing the effect of flow distance on our results, we used 100m, 200m, 500m and infinity as boundaries (see L503 and Figure S24). Accordingly, this lower limit has no relevance for the final results.

**L303: I think a reference to Beven and Kirkby (1979) should be given where you explain the TWI, not Tarboton 1997.**

We will adapt this.

**L329: What model output data did you use for the regression? The median of the MC simulations per catchment? Results from all simulations? […]**

For finding the best explanatory variables, we used the median of the MC simulations per catchment. For the extrapolation to the national scale we used the results from all MC simulations.

We will adapt the manuscript as follows:

[329-331]: "We created a linear regression between each of those catchment statistics to the median fractions of agricultural areas directly, indirectly, and not connected to surface waters, as reported by the LSCM ($f_{LSCM,dir}$, $f_{LSCM,indir}$, $f_{LSCM,nc}$)."

**L329 (cont.): Here I must say that I found using the NECM as an explanatory variable somewhat strange. If I understand this right, you are fitting a linear model to predict the outcomes of your model (LSCM), based on the results of another model (NECM). But if the latter is such a good predictor of runoff connectivity, couldn't you just**

**recommend using it at national scale? At least until you have enough data to parameterize the LSCM for all of Switzerland?**

We indeed think that the NECM is a good (but not a perfect) predictor of runoff connectivity. However, how the NECM performs in describing our data could only be evaluated in retrospect. A priori we did not know that nor did we know whether including other variables could improve on the predictions. Furthermore, our results demonstrate that we have to transform the NECM data to better represent our observations. In summary, the value of the NECM and its limitations for predicting connectivity based on our observations could only be evaluated by building an independent model for comparison.

The NECM model is currently used in practice (e.g. by farmers and local authorities) and we still recommend to continue using it, since it can a) be used for pinpointing critical source areas within a catchment, and b) is a good predictor for connectivity risk in relative terms (e.g. which catchments have a high risk for indirect connectivity). However, looking at connectivity in absolute terms (e.g. which fraction of a catchment is connected directly) our model improves the predictions of the NECM by using additional information from the data we gathered in our 20 study catchments. (Note: We think that the NECM and the NSCM are in fact different from each other and show this in our comment to L547-551.)

In order to parameterize the LSCM for all of Switzerland we would need a map with shortcut locations for the whole country. We don't expect that such a map will be available in near future. Therefore, we had to rely on other nationally available data.

**L355: Is meadow the correct term here?**

After checking again with a native speaker, this includes both, meadows and pasture. We will adapt this accordingly, and use the term "meadows/pasture".

**L436-437: How?**

The directly, indirectly and not connected areas were simply a result of the MC analysis using the surface runoff connectivity model. We suggest to replace this sentence by:

[436-437]: "From the Monte Carlo analysis of the surface runoff connectivity model, we obtained an estimate for the fractions of agricultural areas that are connected directly, indirectly, or not at all to surface waters."

**L443: Here I had the impression you are changing the language with which you describe negative ("While certain areas change their classification: : :") or positive ("for other parts results are very consistent)" results. Please consider rephrasing.**

We are not 100 % sure if we understood this comment correctly. But we suggest to rephrase this to:

[L443:] "The classification of certain catchment parts is changing depending on the model parametrisation (e.g. letters A to C). However, for other parts, the results are consistent across the different MC simulations (e.g. letters D to F)."

**L443 (cont.): Anyway, I found these results quite interesting. Would it be a good idea to look at where the model is consistent and where it is not? I mean, considering model uncertainty, which fields are consistently identified as highly connected? Could these areas be regarded at higher risk of pollutant transport than others? Moreover, if you find out where the models are inconsistent, you can try to figure out why?**

Areas that in reality pose a high risk for surface runoff are not necessarily consistently classified as connected by our model. Whether a high-risk area is consistently classified as connected by the model, depends strongly on how well the DEM is able to represent the responsible flow path, given the DEM uncertainty (resolution and elevation errors) and the uncertainty of infiltration processes (e.g. which flow length leads to infiltration in a forest). Flow paths that depend on the coarse terrain structure and on large landscape features, are represented well (and the connectivity classification is very consistent). However, flow paths that depend on microtopography or small scale landscape features, underlie larger uncertainties (and the connectivity classification is less consistent).

As an example, you could imagine a field close to a river with a steep slope towards the river and without any protection by forests or hedges. The DEM is easily able to reproduce the flow path in this case and the field will be consistently classified as a high-risk area.

Another example could be the case of a field with a steep slope towards a narrow drained farm track. A small ridge along the center of the road stops the water from crossing the road and drains it into the next inlet. This makes the close-by field a high-risk area. However, depending on the model parametrisation, the model will not be able to reproduce the draining effect of the road, since the DEM resolution is too small. Accordingly, this field will not consistently be classified as a high-risk area in this case.

As a consequence, the model can be used to identify some higher-risk areas (which are the areas which are consistently classified as connected). However, many areas that are not consistently classified as connected may pose a similar or higher risk. In this context, it is also important to say that also other factors besides connectivity have an influence of the risk posed by a certain area.

**Figure 5: I didn't understand the colour-ramp bars in the figure legend. Are they necessary?**

When using a D-infinity flow algorithm, the upslope dependence of a raster cell is not classified as "dependent" or "independent", but receives a certain probability to be drained into the receiving area (see also L269-270). E.g. the model MC28 shows some orange areas west-southwest of the letter B. For this model realisation, these areas are indirectly connected with a certain probability ($0 < p_{direct} < 1$) at the same time not connected with a certain probability ($0 < p_{notconnected} < 1$). The sum of the probabilities ($p_{direct}$, $p_{indirect}$ and $p_{notconnected}$) equals 1.

**L547-551: If you propagate the uncertainty in your linear model (e.g. by simulating posteriors of the slope and the intercept and then bootstrapping model predictions), it is likely that these differences will be within your error bands. I guess my question here is: is your extrapolated national model sufficiently different from the NECM to justify its usefulness?**

We agree on this comment and think that the bootstrapping approach is a good approach for addressing the uncertainty. We therefore bootstrapped our linear model and calculated the distribution of the mean of the fractions reported by the NSCM. Since the approach uses a lot of computer capacity we could only run our model for 50 random samples so far, but we will increase the number of random samples to 100. The figure below shows distribution of the mean area fractions resulting from the bootstrapping approach. (Same plot as in Figure S27, but showing the distribution of bootstrapped means.) These preliminary results show that the fractions of directly and not cot connected crop areas reported by the NSCM are significantly different from the results reported by the NECM.

[Figure]

**Figure S28: Area fractions reported by the NECM and distribution of the bootstrapped mean area fractions reported by the NSCM. Directly, indirectly, and not connected crop areas per total agricultural area, non-cropping area per total agricultural area, and indirectly connected crop area per total connected crop area for all catchments in Switzerland. The red dots report the means reported by the NSCM without using a bootstrapping approach.**

We suggest to update the figure above, as soon as the bootstrapping results are available, and to include it as Figure S28 in the manuscript. Additionally, we suggest to modify the manuscript as follows:

[L338-339, additional sentence]: In order to address the uncertainty introduced by the selection of our study catchments, we combined this model fit with a bootstrapping approach.

[L1050 (SI), additional sentence]: To address the uncertainty introduced by the selection of our study catchments, we bootstrapped our model 100 times. For each of the bootstrapping iterations 20 of our study catchments were resampled randomly.

In addition, the numbers reported in L547-551 will be updated based on the results of the bootstrapping approach and a reference to Figure S28 will be added.

Figure 3 will be updated as follows:

[Figure]

**Figure 3: Extrapolation of the local surface runoff connectivity model (LSCM) to the national scale (NSCM) using a** 341 **unit simplex transformation approach.**

**L652: I am curious: which kind of sink filling algorithm would you recommend in this case?**

We don't think that there is a single sink filling algorithm that can deal with this problem in general. In our model, we incorporated our process understanding and our knowledge from field observations to come up with a sink filling algorithm that seems realistic to us (see also L280-282). Additionally, the sensitivity analysis provided insight into how the choice of the sink depth parameter influences our results. For other projects, we would recommend a similar procedure adapted to the question to be answered and meaningful for the topographic characteristics of the landscape.

**L693-697: Here you explain the improvements of the NSCM over the NECM regarding the representation of runoff connectivity, which was helpful. While I agree that the information on crop statistics might help your model, I do not see how the national map incorporates the advantages of the LSCM (i.e. all the impressive field data you collected). In the end I have the impression that this upscaling doesn't do justice to all the work you went through in the small catchments. Moreover, while you appropriately**

**represent the uncertainty of the LSCM, this is somewhat neglected in the extrapolation to the national scale. Would you not expect greater errors in the NSCM than in the LSCM?**

We agree that the NSCM is not able to incorporate the main advantage of the LSCM, which is the availability of field data on shortcut locations. Obviously, we expect larger errors for the NSCM than the LSCM, since the NSCM is simply an extrapolation lacking additional field data besides the data from the 20 catchments. However, since the empirical data used for the NECM were extremely sparse, we wanted to use the results of our field study to come up with an adapted version of this national model based on our field data. We think that this modelling step is a scientifically important step independent of its result. If the resulting NSCM were very similar to the NECM, this would give additional validity to the existing NECM. If the outcome were different, this would give insight whether it over- or underestimates the actual connectivity. This cannot be known a priori but needs the actual comparison.

In our case, we concluded that the models are similar with respect to the areas reported to be indirectly connected. However, the differences reported for the directly connected areas (and also the not connected areas) are quite substantial. For the fractions of directly connected areas, the median of the NSCM corresponds to approximately the 75 % quantile reported in the NECM (see Figure S27).

We agree that the uncertainty was not included sufficiently in the extrapolation to the national scale. We addressed this partially by running our extrapolation model for each of the 100 Monte Carlo runs of the LSCM (see L1049-1050 of the SI). However, we agree that we did not address the uncertainty introduced by the catchment selection. We would like to address this issue by including the results of the bootstrapping approach that you suggested in L547-551.

**References**

Frey, M. P., Schneider, M. K., Dietzel, A., Reichert, P., and Stamm, C.: Predicting critical source areas for diffuse herbicide losses to surface waters: Role of connectivity and boundary conditions, J Hydrol, 365, 23-36, 2009.

Mutzner, L., Bohren, C., Mangold, S., Bloem, S., and Ort, C.: Spatial Differences among Micropollutants in Sewer Overflows: A Multisite Analysis Using Passive Samplers, Environ Sci Technol, 54, 6584-6593, 10.1021/acs.est.9b05148, 2020.

---

## Referee Comment (RC2) · Anonymous Referee #2 · 17 Nov 2020

The paper provides an approach to assess the presence of hydraulic shortcuts on arable land. As the authors admit, the high prevalence of hydraulic shortcuts caused by field and road drainage systems is somewhat specific for Switzerland. Nevertheless, the approaches presented may provide insightful inspiration for researchers with similar questions. The combination of topographic source area modelling and a connectivity model is to my knowledge new and well adapted to the specific conditions and data availability. The model conceptualizes many assumptions about recipient areas, shortcuts and land cover specific responses. Data acquisition and calculations are described in great detail. My major point of criticism concerns the complete lag of calibration / validation for realistic model predictions and parameter uncertainty. This fact is

briefly acknowledged in the discussion but should be explored in the 'further research' section. E.g. by suggestion of appropriate monitoring or deterministic modelling studies. In the same light, the upscaling to national level seems premature. The value of a national map of directly / indirectly connected areas remains unclear without knowing, whether elevated connected proportions correspond to higher or faster hydrological response and respectively increased pesticide wash-off. Given the longer than average extent of the draft, the authors may consider splitting the paper in two parts and adding a more comprehensive risk assessment for the national scale, that also includes pesticide application, hydrologic conditions (see specific comments, below). For the sake of conciseness, some of the definatory and data source specific considerations could have been documented in a separate technical report. References and supplementary material are appropriate and sufficient.

Specific comments:

(1) Lines 106 ff: Shortcut definition should be moved to section 2.2 Assessment of hydraulic shortcuts

(2) Lines 125 ff: The probability of selection was proportional to the total area of arable land. . . How was this represented in the random selection?

(3) Lines 156 ff: How did you prevent selection bias due to drainage plans? I.e. how did you rule out, that no available drainage plan did not correspond to no existing drainage system

(4) Line 243: Elaborate under which circumstances hedge infiltration may be active or inactive

(5) Line 271 ff: Conceptually, the parameter "maximal flow distance" should depend on soil properties and cultivation phase, was this considered?

(6) Line 283: It is unclear how "all possible flow distances were evaluated" with 100 model realizations

(7) Line 295: Suggested section title change to 'hydrological boundary conditions'

(8) Line 312 ff: Although crop type and rain intensity probably don't differ systematically within one catchment, they do impact runoff generation. Should this not be systematically evaluated e.g. across catchments?

(9) Line 384 ff / Tab. 3: why is the destination of such a large number of drainage structures unknown? The maps (Figs. 5, S 2.2.1) suggest line / network structure for most inlets, was the outlet of these unclear? How were unknown drainage locations treated in the connectedness classification?

(10) Lines 419 ff: In Lines 147 ff the field survey was described as "we walked along roads and paths and mapped all the potential shortcut structures." How was mapping accuracy of inlets (5%) and manholes (25.5%) on fields and other areas validated? Lower accuracy esp. in fields with dense vegetation seems likely. How are false positives from mapping ruled out to be false negatives (overlooked structures) from the field survey? How are false negatives from the aerial images quantified altogether?

(11) Line 463: Are the 21 % in Müswangen and 97 % in Boncourt medians / means of the MC ensemble results?

(12) Lines 504 ff: Apart from distribution similarity, how do wetness index and slope affect connected area proportions? I.e. do catchments with higher "hydrological activity" exhibit higher connected proportions?

(13) Lines 515 ff: I suggest to quantify the deviation between NSCM and LSCM with RSME or another goodness of fit measure.

(14) In Fig 7 the mean fraction not connected of LSCM appears to be roughly 15% higher than of NSCM but the text states it is 3% larger (l. 545), why do text and figure differ?

(15) Lines 554 ff: I disagree, that "map corresponds to a risk map of pesticide transport via hydraulic shortcuts". Although hydraulic shortcuts contribute to the risk, surface

runoff proportion and volume, pesticide application intensity and retention in treatment facilities contribute to this risk as well. This is acknowledged in the discussion of NSCM (Lines 700 ff). (See also general comment above)

(16) Line 599: phrase starting with "In Buchs,..." is unclear.

(17) Line 633 – 687 the parameter discussion is somewhat self-referential without calibration data: Road carving depth, sink depth, and shortcut definition should be evaluated / calibrated by observed events. Assumptions such as 'higher DEM resolution is better' or 'manhole sinks should not be filled completely' seem plausible but can't be substantiated by the results of this research. The impact of hydrological activity parameters may not differ between directly and indirectly connected areas of the same catchment, but among catchments and should be evaluated accordingly.

(18) Line 733: Suggested extension: End of pipe measures at shortcut / pipe outlets (treatment, sedimentation, filtration)

(19) In my view, the 2nd research question (line 104) can't be fully answered with the present approach. It should be rephrased and / or referred in the conclusions section.

(20) Line 1116: the table title is confusing. I assume directly connected local surface connectivity model fraction was correlated to directly connected national erosion connectivity model fraction and so on...

––––––––––––––––––––––––

---

## Author Comment (AC2) · 23 Nov 2020

**Answer to Anonymous Referee #2**

We would like to thank Referee #2 for the valuable feedback that highlights some additional points for improving the manuscript. We agree with most of the points brought up and will answer first the general comments, and afterwards the specific comments.

We agree with the major point of criticism concerning the lack of calibration/validation of our model. The Editor also brought up this point. Upon the advice of Referee #2 we suggest to extend the last paragraph of the "further research" section (L750) as follows:

"Although topography-based model estimations can give insight of pesticide transport via hydraulic shortcuts on a large scale, they have not been tested and validated in the field with measurements on flow and transport. Targeted measurements on surface runoff and on pesticide transport through shortcuts are needed to provide evidence on the quantitative relevance of this flow path. A field study in one catchment in the Swiss plateau (Schönenberger, in prep.) demonstrates that pesticide concentrations in shortcuts can be very high. However, more systematic research is needed to quantify the relevance of shortcuts. Ideally, catchment-scale experiments – e.g., with controlled pesticide applications (see Leu et al. (2004);Doppler et al. (2012)) – would be carried out to quantify loss rates from directly and indirectly connected fields. Apart from the practical problems of implementing such experiments in the context of farmers managing their land, this approach will often face the problem that many fields are also tile drained. Consequently, any signal in the stream is a superposition of different potential flow pathways. Given that the transport through shortcuts has no unique characteristic, it is difficult to disentangle and quantify these pathways. This implies that one has to observe simultaneously flow and transport within a catchment at locations where one can differentiate between the flow paths. Such a setup would allow to determine the proportion of total catchment runoff and pesticide load that is transported via hydraulic shortcuts. In addition, isotopic tracers and runoff separation techniques could be used to determine the total amount of surface runoff contributing to catchment runoff. Knowing both, total and indirect surface runoff, the amount of direct surface runoff could be calculated. These measurements of direct and indirect surface runoff could then be compared to the data provided from our connectivity maps. Since shortcuts are only active during rain-events, we suggest an event-based monitoring approach."

Another point of criticism states that "value of a national map of directly / indirectly connected areas remains unclear without knowing, whether elevated connected proportions correspond to higher or faster hydrological response and respectively increased pesticide wash-off" and that therefore the upscaling to the national level seems premature. Also Reviewer #1 stated this part to be scientifically less interesting.

We agree that we did not make it very clear what the added scientific value of this part remains is, given that there is already a similar model on the national scale. We noticed for example that this part was not introduced as one of the objectives of the paper (L102 ff.). The comments by both reviewers clearly demonstrate that we need to sharpen our arguments and describe concisely what the purpose of that part of the paper is and to eliminate those parts in the manuscript that do not fit the specific objective.

The rationale for the development of our map at the national scale is that there is first a need for connectivity data at that scale (e.g. for people evaluating the Swiss Action Plan on pesticides) and that they rely on the existing connectivity map. Given that we substantially improved on the empirical basis at the catchment scale regarding connectivity, we consider it scientifically justified to evaluate how our findings affect predictions at the catchment and the

national scale. Actually, we consider this a necessity taking the societal role of science serious: not providing such a comparison would imply that we demonstrate with our empirical observations that connectivity indeed matters according the best knowledge we have, but leaving it to others (who have no better means than we have) to figure out what our findings mean in the larger context. Hence, what we do with the upscaling step is to bring together existing scientific knowledge in a transparent manner.

Reviewer 2 asked the question whether or not the proposition of a connectivity map at the national scale wasn't premature given the lack of empirical validation of the model. This is a very well founded question that has to be considered seriously. However, this question does not only address our map at the large scale but also the use of the already published map, which is intensively used in practice. What does a statement of prematureness in this context mean? If one accepts that one should use the best available knowledge for rationale decision making, the question is whether one should make use of such maps at the current stage or whether their use does more harm than providing benefits. Causing more harm than benefits could be called a premature use of a model from a decision-making perspective.

Based on the comparison of our empirical data, the existing model and our "new" model, we can conclude that both models can reasonably well represent the field observations. We don't have any indications that would suggest that it is preferable not to use any of the large-scale models for assessing the connectivity of fields to surface water bodies. In this sense, we conclude that is not premature to use either of these models. Therefore, it also makes sense to evaluate how our observations affect the existing model and which differences are induced.

This argument does of course not invalidate the correct critique that neither of the two models have been tested and validated empirically regarding their actual capacity to quantify the connectivity effects on water flow and transport of agrochemicals beyond the few observational studies that triggered this kind of connectivity assessment in the first place. This aspect needs to be clearly communicated.

In summary, we propose the following modifications of the paper to address the concerns of Reviewer 2 (these changes partially also address the issues raised by the first reviewer):

- The objective of evaluating the consequences are clearly stated in the Introduction and an explicit rationale for doing so is provided (How do the additional empirical observation influence the large-scale predictions?).
- The limitations of existing large-scale models are clearly stated, the research need is emphasised, and possible approaches explicitly discussed.
- The focus of the large-scale aspect is on the comparison with the existing model. To be consistent with this objective, we replace the current Fig. 8 with a map depicting the differences between the two models. The focus will be clearly on how the additional empirical observations influence the model predictions.
- Along the same lines, we will skip those parts of the text that discuss the findings specifically for the pesticide issues (L552-561).

**Specific comments**

**(1) Lines 106 ff: Shortcut definition should be moved to section 2.2 Assessment of hydraulic shortcuts:**

We will move this.

**(2) Lines 125 ff: The probability of selection was proportional to the total area of arable land...How was this represented in the random selection?**

We are not sure if we understand the question correctly, but try to answer it as good as possible. We performed a weighted random selection from a list of all catchments. The weights of each catchment equalled the total area of arable land in the catchment. Specifically, we were using the python function "numpy.random.choice":

```
numpy.random.choice(a = catchment_id_list, size = 20, p = catchment_area_list)
```

**(3) Lines 156 ff: How did you prevent selection bias due to drainage plans? I.e. how did you rule out, that no available drainage plan did not correspond to no existing drainage system**

We tried to reduce the impact of selection bias as much as possible by using three different acquisition methods. If drainage plans are not available in a certain catchment, the other two methods are to some degree filling this gap and accordingly reducing the selection bias. As shown in Table 5, the drainage plan mapping method had a lower recall than the aerial image mapping methods. Additionally, also the number of shortcuts identified by aerial images is much higher than by drainage plans. Therefore, the aerial image mapping method is more important for the overall result and we expect the selection bias due to drainage plans to be small. However, we are sure that we still missed some of the shortcuts and addressed this in L424-226 by writing that the numbers reported are a lower boundary estimate.

**(4) Line 243: Elaborate under which circumstances hedge infiltration may be active or inactive**

We could imagine various factors affecting the runoff capturing efficiency of a hedge. For example, the width of the hedge, shrub species, or the degree of runoff concentration. We did choose this parameter to have a binary distribution since no further information on the hedges (such as hedge width) were available. As shown in the sensitivity analysis (Figure S21 and S22), the hedge width parameter only has a minor influence on the overall results. Therefore, we also did not spend time in refining this parameter further.

**(5) Line 271 ff: Conceptually, the parameter "maximal flow distance" should depend on soil properties and cultivation phase, was this considered?**

This was not considered directly, but indirectly. Since no data on soil properties is available on national scale in Switzerland, we tried to identify potential differences in soil properties by calculating the topographic wetness index (TWI) (see L295 ff.). We did not find any systematic differences between the TWI distributions of directly and indirectly connected areas (see L508 ff.). We therefore also do not expect systematic differences in soil properties

between directly and indirectly connected areas. We did not address the cultivation phase specifically. This influence factor is expected to cause large differences in maximal flow distances in time and space. However, we again do not expect systematic differences of this influence factor between directly and indirectly connected areas.

Consequently, we expect that the maximal flow distances found on directly and indirectly connected areas are not systematically different from each other.

**(6) Line 283: It is unclear how "all possible flow distances were evaluated" with 100 model realizations**

We agree that this is not clear. We will change this to:

"For the parameter maximal flow distance, all possible flow distances were evaluated for each Monte Carlo simulation."

**(7) Line 295: Suggested section title change to 'hydrological boundary conditions'**

We think the more specific term "hydrological activity" fits better here, since this term is usually used in the context of critical source areas, for example see: Pionke et al. (2000)

**(8) Line 312 ff.: Although crop type and rain intensity probably don't differ systematically within one catchment, they do impact runoff generation. Should this not be systematically evaluated e.g. across catchments?**

Yes, these factors affect runoff generation and are expected to differ systematically between catchments. As also mentioned in L299-301, this manuscript focuses on comparing indirect surface runoff to direct surface runoff. We therefore were looking for systematic differences between indirectly and directly connected areas. Comparing surface runoff generation across catchments was not within the scope of this manuscript. Currently, except from rainfall data, the data availability for such a comparison is not given. For example, crop data are currently not available in sufficient resolution (see also L349-350), soil maps are not available on a national scale. It is also unknown how farming practices (e.g. pesticide application or soil management) differ between catchments.

We however agree that this could be an interesting direction to go and suggest this in the "further research" section.

**(9) Line 384 ff / Tab. 3: why is the destination of such a large number of drainage structures unknown? The maps (Figs. 5, S 2.2.1) suggest line / network structure for most inlets, was the outlet of these unclear? How were unknown drainage locations treated in the connectedness classification?**

Three reasons were mainly responsible for this problem:

1) There was no drainage plan available in the whole catchment.
2) Drainage plans were available in the catchment, but did not cover the specific region where the potential shortcut was located.
3) Drainage plans were available in the catchment and did cover the specific region where the potential shortcut was located, but the potential shortcut and its drainage

structure were not shown on the plans. (They were however identified during the field survey or on the aerial images.)

For the inlets with known drainage locations 99 % were connected to the surface waters (87 %) or via WWTP/CSO (12 %). Therefore, we assumed in the connectivity model that all shortcuts with unknown drainage locations drain to surface waters (see L252-255).

**(10) Lines 419 ff: In Lines 147 ff the field survey was described as "we walked along roads and paths and mapped all the potential shortcut structures." How was mapping accuracy of inlets (5%) and manholes (25.5%) on fields and other areas validated? Lower accuracy esp. in fields with dense vegetation seems likely.**

The accuracy of field mapping could not be validated since we would need a "ground truth" for this, which was not available. We agree that lower accuracy in fields with dense vegetation is likely and discussed this issue in L608-612.

**(10 cont.) How are false positives from mapping ruled out to be false negatives (overlooked structures) from the field survey? How are false negatives from the aerial images quantified altogether?**

From your question, we noticed that Table 5 is not clear enough. As you state correctly, we cannot differentiate between false positives from the aerial image/drainage plan method and false negatives (=overlooked structures) from the field survey when looking at the *identification* of a shortcut structure. For the *identification,* we therefore only report the recall. However, given that a shortcut structure is identified, we can analyse if it is *classified* correctly by the aerial image/drainage plan method (see also L414-418). For the *classification,* we can also report false negatives. For example, a shortcut structure was identified by the aerial image method and was classified as a maintenance manhole. In fact, the field survey showed that the structure showed that the structure is an inlet. This would correspond to a false negative classification.

We will adapt Table 5 as follows, to make the difference between identification and classification accuracies more clear.

Table 1: Recall and classification accuracies of the mapping methods aerial images and drainage plans. The recall corresponds to the probability that a potential shortcut is found by the mapping method. Percentages indicate the recall of each individual mapping method. In brackets, the recall of the combination of both methods is given. The accuracy corresponds to the sum of true positive fraction and true negative fraction.

| Mapping method | Manhole type | Identification | Classification | | | | |
|---|---|---|---|---|---|---|---|
| | | Recall | True positives | False positives | True negatives | False negatives | Accuracy |
| Aerial images | Inlets | 53 % (60 %) | 61 % | 1.3 % | 33 % | 4.9 % | 94 % |
| | Maintenance manholes | 62 % (69 %) | 32 % | 5.3 % | 61 % | 1.3 % | 93 % |
| Drainage plans | Inlets | 32 % (60 %) | 67 % | 4.5 % | 22 % | 6.6 % | 89 % |
| | Maintenance manholes | 21 % (69 %) | 20 % | 7.1 % | 68 % | 5.3 % | 88 % |

**(11) Line 463: Are the 21 % in Müswangen and 97 % in Boncourt medians / means of the MC ensemble results?**

Yes, those are the medians. We will adapt the sentence as follows:

"However, this fraction varies strongly between the study areas, with median fractions ranging from 21 % in Müswangen to 97 % in Boncourt."

**(12) Lines 504 ff: Apart from distribution similarity, how do wetness index and slope affect connected area proportions? I.e. do catchments with higher "hydrological activity" exhibit higher connected proportions?**

Our connectivity model produces something like a "theoretical connectivity map", i.e. the areas reported are connected under the assumption that surface runoff is produced and that this surface runoff is not infiltrating before reaching the recipient area. In contrast, the "effective connectivity" depends on the amount of surface runoff produced and the amount of surface runoff infiltrating before reaching the recipient area.

Wetness index and slope are positively correlated to the probability of an area to be hydrologically active, i.e. more surface runoff is produced. Additionally, they are negatively correlated to the probability that surface runoff infiltrating before reaching the recipient area.

Accordingly, areas with higher wetness index and slope are expected to exhibit a higher "effective connectivity". In fact, those considerations were the reason for performing the analysis described in L504 ff.

**(13) Lines 515 ff: I suggest to quantify the deviation between NSCM and LSCM with RSME or another goodness of fit measure.**

We calculated the RSME between the NECM and the LSCM, and between the NSCM and the LSCM. We will adapt L520-523 as follows:

"The differences to the LSCM were strongly reduced by this transformation. The root-mean-square error (RSME) reduced from 17 % to 9.5 % for directly connected fractions, from 12 % to 7.6 % for indirectly connected fractions, and from 18 % to 7.6 % for not connected fractions."

**(14) In Fig 7 the mean fraction not connected of LSCM appears to be roughly 15% higher than of NSCM but the text states it is 3% larger (l. 545), why do text and figure differ?**

There is an error in the legend of Figure 7. While the colours were described correctly in the figure description, in the legend (on the top right of the figure) the colours "red" and "blue" were interchanged. We corrected Figure 7 (see below) and will adapt it in the manuscript.

[Figure]

**Figure 7: Fractions of directly connected (f_dir), indirectly connected (f_indir), and not connected areas (f_nc) per total agricultural area for the local surface runoff connectivity model (LSCM, blue), national erosion connectivity model (NECM, red), and national surface runoff connectivity model (NSCM, green) in the 20 study areas. Small blue circles represent the catchment medians of all Monte Carlo simulations of the LSCM, small red circles represent the data reported by the NECM, and small green circles represent the catchment medians of the NSCM. Large circles represent the means of the LSCM (blue), NECM (red), and NSCM data (green). Shaded areas represent normal Kernel density estimates of the LSCM, NECM, and NSCM data.**

**(15) Lines 554 ff: I disagree, that "map corresponds to a risk map of pesticide transport via hydraulic shortcuts". Although hydraulic shortcuts contribute to the risk, surface runoff proportion and volume, pesticide application intensity and retention in treatment facilities contribute to this risk as well. This is acknowledged in the discussion of NSCM (Lines 700 ff). (See also general comment above)**

We agree with this point. With revising and shortening the second part of the paper (as written in our answer to the general comments), we will replace Figure 8 and this sentence.

**(16) Line 599: phrase starting with "In Buchs,..." is unclear.**

We will rephrase this to:

"In Buchs, around 60 % of the channel drain and ditch length consists of ditches that cannot be clearly distinguished from small streams."

**(17) Line 633 – 687 the parameter discussion is somewhat self-referential without cal-ibration data: Road carving depth, sink depth, and shortcut definition should be evaluated / calibrated by observed events. Assumptions such as 'higher DEM resolution is better' or 'manhole sinks should not be filled completely' seem plausible**

**but can't be substantiated by the results of this research. The impact of hydrological activity parameters may not differ between directly and indirectly connected areas of the same catchment, but among catchments and should be evaluated accordingly.**

We agree that our quantitative results cannot substantiate statements that higher DEM resolution was better. However, the field observations – e.g. based on visual inspections of sediment deposition along roads – clearly revealed that the microtopography can play a major role in controlling the flux of water, solutes and sediments into shortcuts. Furthermore, one has to be aware that these parameters are global parameters in the model despite the fact that there might be regional differences. The optimal road carving depth for example may differ according to topography, regional construction standards, etc. Given this situation, we also think that the optimal way to go would be to calibrate these parameters by observed events. However, on a scale of 20 catchments this would be an extremely laborious task. Additionally, field observations of these parameters also underlie high uncertainties. For example, sink depths are strongly variable in time and space. We therefore aimed on discussing other options that could be used to improve our results. To clarify that these statements are not findings that can be substantiated by our results, but a discussion of improvement options, we will modify the manuscript as follows:

- Higher DEM resolution: We will replace the word "would" in the sentence in L637-638 by the word "could", since we can't tell from our results that this would really improve the model.
- Sink filling: Similarily, we will replace the word "can" by the word "could" in L650-651.

**(18) Line 733: Suggested extension: End of pipe measures at shortcut / pipe outlets(treatment, sedimentation, filtration)**

We agree on this and will adapt this sentence to:

"Other measures could aim on the shortcut structures themselves (e.g. construction of shortcuts as small infiltration basins, removal of shortcuts, or treatment of water in shortcuts) or on the pipe outlets (e.g. drainage of shortcuts to infiltration basins, treatment of water at the pipe outlet)."

**(19) In my view, the 2nd research question (line 104) can't be fully answered with the present approach. It should be rephrased and / or referred in the conclusions section.**

In our view, we can actually answer the second research question with our approach. We did not find any evidence on systematic differences in hydrological activity. In addition, we do not expect systematic differences in farming practices, precipitation, or crop types between directly and indirectly connected areas. Given the current knowledge, we therefore expect that the proportions of direct and indirect surface runoff related pesticide transport are proportional to the directly and indirectly connected area. However, we think that this is not formulated clear enough and we will revise the conclusions (L758) and results (L512) accordingly.

**(20) Line 1116: the table title is confusing. I assume directly connected local surface connectivity model fraction was correlated to directly connected national erosion connectivity model fraction and so on...**

We agree and will adapt the table accordingly (see below).

**Table S 1: Correlation of catchment statistics with fractions of connected area connectivity. NECM: National erosion connectivity model, LSCM: Local surface runoff connectivity model.**

| Variable | Fraction directly connected $f_{LSCM,dir}$ (-) | | | Fraction indirectly connected $f_{LSCM,indir}$ (-) | | | Fraction not connected $f_{LSCM,nc}$ (-) | | |
|---|---|---|---|---|---|---|---|---|---|
| | $R^2$ | Slope | P | $R^2$ | Slope | P | $R^2$ | Slope | P |
| NECM: Directly connected agricultural area per total agricultural area $f_{NECM,dir}$ (-) | 0.71 | 1.0E+00 | < 0.001 *** | | | | | | |
| NECM: Indirectly connected agricultural area per total agricultural area $f_{NECM,indir}$ (-) | | | | 0.52 | 6.0E-01 | < 0.001 *** | | | |
| NECM: Not connected agricultural area per total agricultural area $f_{NECM,nc}$ (-) | | | | | | | 0.26 | 4.0E-01 | 0.022 * |
| Surface water body density (m⁻¹) | 0.51 | 2.2E+02 | < 0.001 *** | 0.35 | -1.4E+02 | 0.006 ** | 0.14 | -7.6E+01 | 0.10 * |
| Paved road density (m⁻¹) | 0.20 | -2.2E+01 | 0.049 * | 0.19 | 1.7E+01 | 0.053 - | 0.04 | 6.5E+00 | 0.41 - |
| Inlet density (ha⁻¹) | 0.07 | -1.3E-01 | 0.28 - | 0.10 | 1.2E-01 | 0.17 - | 0.00 | 1.0E-02 | 0.90 - |
| Manhole density (ha⁻¹) | 0.15 | 4.0E+02 | 0.09 - | 0.07 | -2.0E+02 | 0.27 - | 0.07 | -1.8E+02 | 0.27 - |
| Yearly rainfall (mm/year) | 0.10 | -5.2E-02 | 0.17 - | 0.06 | 3.2E-02 | 0.28 - | 0.04 | 2.0E-02 | 0.43 - |
| Total road density (m⁻¹) | 0.05 | 2.6E-01 | 0.35 - | 0.05 | -2.0E-01 | 0.33 - | 0.00 | -4.5E-02 | 0.80 - |
| Subsurface waterbody density (m⁻¹) | 0.11 | -7.5E+00 | 0.14 - | 0.04 | 3.3E+00 | 0.40 - | 0.10 | 4.5E+00 | 0.18 - |
| Fraction of agricultural area (-) | 0.00 | 2.6E+01 | 0.94 - | 0.03 | -1.7E+02 | 0.48 - | 0.03 | 1.7E+02 | 0.43 - |
| Unpaved road density (m⁻¹) | 0.15 | 4.4E-04 | 0.09 - | 0.02 | -1.2E-04 | 0.55 - | 0.18 | -3.2E-04 | 0.063 - |
| Lake shore density (m⁻¹) | 0.03 | 1.3E-02 | 0.49 - | 0.02 | 7.7E-03 | 0.60 - | 0.13 | -1.9E-02 | 0.13 - |
| Slope on agricultural areas (°) | 0.04 | -5.8E+00 | 0.41 - | 0.00 | 2.2E-01 | 0.97 - | 0.09 | 6.0E+00 | 0.19 - |

**References**

Doppler, T., Camenzuli, L., Hirzel, G., Krauss, M., Lück, A., and Stamm, C.: Spatial variability of herbicide mobilisation and transport at catchment scale: insights from a field experiment, Hydrol Earth Syst Sc, 16, 1947-1967, https://doi.org/10.5194/hess-16-1947-2012, 2012.

Leu, C., Singer, H., Stamm, C., Müller, S. R., and Schwarzenbach, R. P.: Variability of Herbicide Losses from 13 Fields to Surface Water within a Small Catchment after a Controlled Herbicide Application, Environ Sci Technol, 38, 3835-3841, https://doi.org/10.1021/es0499593, 2004.

Pionke, H. B., Gburek, W. J., and Sharpley, A. N.: Critical source area controls on water quality in an agricultural watershed located in the Chesapeake Basin, Ecol Eng, 14, 325-335, Doi 10.1016/S0925-8574(99)00059-2, 2000.

---

## Author Comment (AC3) · 7 Dec 2020

As pointed out by the editor, there is an error in one sentence of our answer to the Reviewer comment concerning L116-117 of the manuscript. In the following, we provide the corrected answer.

Reviewer comment: I am curious to why shortcuts that drain into surface waters or treatment plants are treated same by the model. If you are looking at pollutant transport, shouldn't there be a difference?

Corrected answer: We agree that there is a difference between these two processes.

In waste water treatment plants pesticides are removed to a certain degree. However, during heavy rain events (i.e. the point in time when the largest pesticide loads are expected in shortcuts) rain water is often not reaching WWTPs but directed to surface waters through CSOs. CSOs have been reported to be an important pesticide transport pathway (Mutzner et al., 2020) and we expect the transport via CSO to be very similar to "normal" shortcuts. In addition, from our field studies we know transport via WWTPs/CSO is less important. Only 12% of the inlets mapped drain to WWTPs/CSO while 87% drain to surface waters (see L382-384). We therefore do not expect these differences to have a major influence on our results and decided to neglect them.

---

## Author Response (AR1)

**Author's response**

In the following, our point-by-point response to the reviews is given. Line numbers of the reviewer comments refer to the old manuscript. All other line numbers refer to the new files: First, line numbers in the new version of the manuscript are given, followed by line numbers in the author's track-changes file ("TC", in brackets). Changes in the manuscript are marked in red.

**Answer to Anonymous Referee #1**

We would like to thank Referee #1 for the constructive and valuable feedback. We mostly agree with the referee's comments and will address them individually in the following. We still think that the manuscript part about the upscaling to the national level is important and scientifically interesting and will explain our arguments in more detail below. Nevertheless, we agree on the criticism brought up concerning to this part and adapted the manuscript to address it. Since concerns regarding the upscaling to the national model were also brought up by Referee #2, we additionally provide a comprehensive discussion on this part in our general answer to Referee #2.

**L42: Please consider rephrasing to: "relevant process have to be understood".**

We agree on this. We rephrased it accordingly. → L44 (TC: L44)

**L116-117: I am curious to why shortcuts that drain into surface waters or treatment plants are treated same by the model. If you are looking at pollutant transport, shouldn't there be a difference?**

We agree that there is a difference between these two processes. In waste water treatment plants pesticides are removed to a certain degree. However, during heavy rain events (i.e. the point in time when the largest pesticide loads are expected in shortcuts) rain water is often not reaching WWTPs but directed to surface waters through CSO. CSOs have been reported to transport pesticides (Mutzner et al., 2020) and we expect the transport via CSO to be very similar to "normal" shortcuts. In addition, from our field studies we know transport via WWTPs/CSO is less important. Only 12 % of the inlets mapped drain to WWTPs/CSO while 87 % drain to surface waters (see L381-383, TC: 395-397). We therefore do not expect these differences to have a major influence on our results and decided to neglect them.

**L182: Please consider rephrasing to: "In order to better understand: : :"**

We agree on this and rephrased it accordingly. → L189 (TC: L191)

**Figure 2: Please define WWTP/CSO in the legend.**

We added a definition for these abbreviations in the legend. → Figure 2 (TC: Figure 2)

**L236: What are internal sinks?**

Internal sinks are depressions from where water cannot flow on the soil surface to a receiving water, but infiltrates water locally. This term is for example also used in Frey et al. (2009).

**L258: Regarding the connectivity model: Maybe I missed something, but how do you go from the upslope dependence output raster to defining if a cell is directly or indirectly connected?**

For every Monte Carlo parameter combination, we ran the tool "D-Infinity upslope dependence" three times. Each time we used a different type of recipient area (surface waters, shortcuts, infiltration areas) as an input. This resulted in three upslope dependence output rasters – one each for each type of recipient areas. These rasters then defined if a cell is directly, indirectly, or not connected.

**L265: How was this "carving" performed? Did you use a stream burning algorithm?**

We carved the recipient areas into the DEM by rasterizing topographic data and lowering the corresponding cells. For the recipient area type "surface waters" this corresponds to a (very simple) stream burning algorithm. We did not impose an additional gradient towards the stream as it is done by some stream burning algorithms. We think that using such a gradient would not change our results, since the river course of the raw DEM and the burnt-in river course align very well.

**L274-275 & Table 2: The 2m limit for the maximal flow distance seems unrealistically low. Is there a reason why you chose this value?**

Yes, we agree that this is unrealistically low. However, for the maximal flow distance we simply calculated all possible values. Since our DEM has a resolution of 2m, this was simply the shortest flow distance that could be calculated. For analysing the effect of flow distance on our results, we used 100m, 200m, 500m and infinity as boundaries (see Figure S24). Accordingly, this lower limit has no relevance for the final results.

This was not written clearly in the manuscript. We therefore adapted the manuscript to make this procedure easier to understand. → L275-278, 463-464 (TC: L282-297, L484-485)

**L303: I think a reference to Beven and Kirkby (1979) should be given where you explain the TWI, not Tarboton 1997.**

We adapted this in the manuscript. → L305 (TC: L312)

**L329: What model output data did you use for the regression? The median of the MC simulations per catchment? Results from all simulations? […]**

For finding the best explanatory variables, we used the median of the MC simulations per catchment. For the extrapolation to the national scale, we used the results from all MC simulations.

We adapted the manuscript as follows:

"We created a linear regression between each of those catchment statistics to the median fractions of agricultural areas directly, indirectly, and not connected to surface waters, as reported by the LSCM ($f_{LSCM,dir}$, $f_{LSCM,indir}$, $f_{LSCM,nc}$)." → L327-329 (TC: L339-341)

**L329 (cont.): Here I must say that I found using the NECM as an explanatory variable somewhat strange. If I understand this right, you are fitting a linear model to predict the outcomes of your model (LSCM), based on the results of another model (NECM). But if the latter is such a good predictor of runoff connectivity, couldn't you just recommend using it at national scale? At least until you have enough data to parameterize the LSCM for all of Switzerland?**

We indeed think that the NECM is a good (but not a perfect) predictor of runoff connectivity (and stated this more clearly in the updated manuscript: L715-716; TC: L765-766). However, how the NECM performs in describing our data could only be evaluated in retrospect. A priori, we did not know that nor did we know whether including other variables could improve on the predictions. Furthermore, our results demonstrate that we have to transform the NECM data to better represent our observations. In summary, the value of the NECM and its limitations for predicting connectivity based on our observations could only be evaluated by building an independent model for comparison.

The NECM model is currently used in practice (e.g. by farmers and local authorities) and we still recommend to continue using it (we stated this more clearly in the manuscript L719-720; TC: L769-771), since it can a) be used for pinpointing critical source areas within a catchment, and b) is a good predictor for connectivity risk in relative terms (e.g. which catchments have a high risk for indirect connectivity). However, looking at connectivity in absolute terms (e.g. which fraction of a catchment is connected directly) our model improves the predictions of the NECM by using additional information from the data we gathered in our 20 study catchments. (Note: Additional analyses showed that the NECM and the NSCM are in fact different from each other (see answer to Reviewer #2 and L553-562, TC: 588-597).)

In order to parameterize the LSCM for all of Switzerland we would need a map with shortcut locations for the whole country. We don't expect that such a map will be available in near future. Therefore, we had to rely on other nationally available data.

Since also the 2nd reviewer critizised this part of the manuscript, we modified it. We are now more clearly recommending to use the NECM at the national scale and mainly focus on the differences between the NECM and the NSCM (see answer to Anonymous Referee #2).

**L355: Is meadow the correct term here?**

After checking again with a native speaker, this includes both, meadows and pasture. We adapted this accordingly, and used the term "meadows/pasture". → e.g. L352 (TC: L364-365)

**L436-437: How?**

The directly, indirectly and not connected areas were simply a result of the MC analysis using the surface runoff connectivity model. We replaced this sentence by:

"From the Monte Carlo analysis of the surface runoff connectivity model, we obtained an estimate for the fractions of agricultural areas that are connected directly, indirectly, or not at all to surface waters." → L435-436 (TC: L450-451)

**L443: Here I had the impression you are changing the language with which you describe negative ("While certain areas change their classification: : :") or positive ("for other parts results are very consistent)" results. Please consider rephrasing.**

We are not 100 % sure if we understood this comment correctly. However, we rephrased this to:

"The classification of certain catchment parts is changing depending on the model parametrisation (e.g. letters A to C). However, for other parts, the results are consistent across the different MC simulations (e.g. letters D to F)." → L440-442 (TC: 457-459)

**L443 (cont.): Anyway, I found these results quite interesting. Would it be a good idea to look at where the model is consistent and where it is not? I mean, considering model uncertainty, which fields are consistently identified as highly connected? Could these areas be regarded at higher risk of pollutant transport than others? Moreover, if you find out where the models are inconsistent, you can try to figure out why?**

Areas that in reality pose a high risk for surface runoff are not necessarily consistently classified as connected by our model. Whether a high-risk area is consistently classified as connected by the model, depends strongly on how well the DEM is able to represent the responsible flow path, given the DEM uncertainty (resolution and elevation errors) and the uncertainty of infiltration processes (e.g. which flow length leads to infiltration in a forest). Flow paths that depend on the coarse terrain structure and on large landscape features, are represented well (and the connectivity classification is very consistent). However, flow paths that depend on microtopography or small scale landscape features, underlie larger uncertainties (and the connectivity classification is less consistent).

As an example, you could imagine a field close to a river with a steep slope towards the river and without any protection by forests or hedges. The DEM is easily able to reproduce the flow path in this case and the field will be consistently classified as a high-risk area.

Another example could be the case of a field with a steep slope towards a narrow drained farm track. A small ridge along the center of the road stops the water from crossing the road and drains it into the next inlet. This makes the close-by field a high-risk area. However, depending on the model parametrisation, the model will not be able to reproduce the draining effect of the road, since the DEM resolution is too small. Accordingly, this field will not consistently be classified as a high-risk area in this case.

As a consequence, the model can be used to identify some higher-risk areas (which are the areas which are consistently classified as connected). However, many areas that are not consistently classified as connected may pose a similar or higher risk. In this context, it is also important to say that also other factors besides connectivity have an influence of the risk posed by a certain area.

**Figure 5: I didn't understand the colour-ramp bars in the figure legend. Are they necessary?**

When using a D-infinity flow algorithm, the upslope dependence of a raster cell is not classified as "dependent" or "independent", but receives a certain probability to be drained

into the receiving area (see also L269-270). E.g. the model MC28 shows some orange areas west-southwest of the letter B. For this model realisation, these areas are indirectly connected with a certain probability ($0 < p_{direct} < 1$) at the same time not connected with a certain probability ($0 < p_{notconnected} < 1$). The sum of the probabilities ($p_{direct}$, $p_{indirect}$ and $p_{notconnected}$) equals 1.

We rephrased the description of Figure 5 to make this more clear.
→ L446-451 (TC: L466-472)

**L547-551: If you propagate the uncertainty in your linear model (e.g. by simulating posteriors of the slope and the intercept and then bootstrapping model predictions), it is likely that these differences will be within your error bands. I guess my question here is: is your extrapolated national model sufficiently different from the NECM to justify its usefulness?**

We agree on this comment and think that the uncertainty in general was not addressed well enough in the NSCM model. We therefore performed the following two additional analyses to address this comment.

1) To address the uncertainty introduced by the catchment selection, we bootstrapped our results 100 times, resampling the 20 catchments.

   The results show (L560-562; TC: L596-597) that the differences between the two models are significantly larger than the uncertainty introduced by the selection of the study catchments.

   We additionally modified the manuscript as follows:

   - Additional sentence (Methods): "To address the uncertainty introduced by the selection of our study catchments, we additionally bootstrapped the model one hundred times." → L336-337 (TC: L348-349)
   - Updated Figure 3 (TC: Figure 3)
   - Additional sentence (SI methods): "To address the uncertainty introduced by the selection of our study catchments, we bootstrapped our model 100 times. For each of the bootstrapping iterations 20 of our study catchments were resampled randomly." → L1111-1113 (TC: L1183-1185)

2) Already in the previous version of the manuscript, the uncertainty of all MC simulations of the LSCM model was propagated to the NSCM model. However, in the results we only reported the mean fractions. How the NSCM is influenced by different model parameters (different MC simulations) was not reported. We now added the 5% and 95% quantiles of the NSCM model to the results section and compared these quantiles to the NESCM model. The results show that some (but not all) of the NSCM connectivity fractions are significantly different from the NECM connectivity fractions.

   Adaptions to the manuscript:

   - Results: L522-523, L542, **L549-562** (TC: L547-548, L577, **L584-595**)
   - Discussion: L705-707 (TC: L755-756)
   - SI: Table S9 (TC: Table S9)

**L652: I am curious: which kind of sink filling algorithm would you recommend in this case?**

We don't think that there is a single sink filling algorithm that can deal with this problem in general. In our model, we incorporated our process understanding and our knowledge from field observations to come up with a sink filling algorithm that seems realistic to us (see also L283-285, TC: 288-290). Additionally, the sensitivity analysis provided insight into how the choice of the sink depth parameter influences our results. For other projects, we would recommend a similar procedure adapted to the question to be answered and meaningful for the topographic characteristics of the landscape.

**L693-697: Here you explain the improvements of the NSCM over the NECM regarding the representation of runoff connectivity, which was helpful. While I agree that the information on crop statistics might help your model, I do not see how the national map incorporates the advantages of the LSCM (i.e. all the impressive field data you collected). In the end I have the impression that this upscaling doesn't do justice to all the work you went through in the small catchments. Moreover, while you appropriately represent the uncertainty of the LSCM, this is somewhat neglected in the extrapolation to the national scale. Would you not expect greater errors in the NSCM than in the LSCM?**

We agree that the NSCM is not able to incorporate the main advantage of the LSCM, which is the availability of field data on shortcut locations. Obviously, we expect larger errors for the NSCM than the LSCM, since the NSCM is simply an extrapolation lacking additional field data besides the data from the 20 catchments. However, since the empirical data used for the NECM were extremely sparse, we wanted to use the results of our field study to come up with an adapted version of this national model based on our field data. We think that this modelling step is a scientifically important step independent of its result. If the resulting NSCM were very similar to the NECM, this would give additional validity to the existing NECM. If the outcome were different, this would give insight whether it over- or underestimates the actual connectivity. This cannot be known a priori but needs the actual comparison.

As already mentioned in our answer to your previous comment (see above; comment to L547-551 of the original manuscript), we agree that the uncertainty was not addressed sufficiently in the extrapolation to the national scale and adapted the manuscript accordingly.

**Answer to Anonymous Referee #2**

We would like to thank Referee #2 for the valuable feedback that highlights some additional points for improving the manuscript. We agree with most of the points brought up and will answer first the general comments, and afterwards the specific comments.

We agree with the major point of criticism concerning the lack of calibration/validation of our model. The Editor also brought up this point. Upon the advice of Referee #2 we extended the last paragraph of the "further research". → L779-799 (TC: 846-866)

Another point of criticism states that "value of a national map of directly / indirectly connected areas remains unclear without knowing, whether elevated connected proportions correspond to higher or faster hydrological response and respectively increased pesticide wash-off" and that therefore the upscaling to the national level seems premature. Also Reviewer #1 stated this part to be scientifically less interesting.

We agree that we did not make it very clear what the added scientific value of this part remains is, given that there is already a similar model on the national scale. We noticed for example that this part was not introduced as one of the objectives of the paper. The comments by both reviewers clearly demonstrate that we need to sharpen our arguments and describe concisely what the purpose of that part of the paper is and to eliminate those parts in the manuscript that do not fit the specific objective.

The rationale for the development of our map at the national scale is that there is first a need for connectivity data at that scale (e.g. for people evaluating the Swiss Action Plan on pesticides) and that they rely on the existing connectivity map. Given that we substantially improved on the empirical basis at the catchment scale regarding connectivity, we consider it scientifically justified to evaluate how our findings affect predictions at the catchment and the national scale. Actually, we consider this a necessity taking the societal role of science serious: not providing such a comparison would imply that we demonstrate with our empirical observations that connectivity indeed matters according the best knowledge we have, but leaving it to others (who have no better means than we have) to figure out what our findings mean in the larger context. Hence, what we do with the upscaling step is to bring together existing scientific knowledge in a transparent manner.

Reviewer 2 asked the question whether or not the proposition of a connectivity map at the national scale wasn't premature given the lack of empirical validation of the model. This is a very well founded question that has to be considered seriously. However, this question does not only address our map at the large scale but also the use of the already published map, which is intensively used in practice. What does a statement of prematureness in this context mean? If one accepts that one should use the best available knowledge for rationale decision making, the question is whether one should make use of such maps at the current stage or whether their use does more harm than providing benefits. Causing more harm than benefits could be called a premature use of a model from a decision-making perspective.

Based on the comparison of our empirical data, the existing model and our "new" model, we can conclude that both models can reasonably well represent the field observations. We don't have any indications that would suggest that it is preferable *not* to use any of the large-scale models for assessing the connectivity of fields to surface water bodies. In this sense, we conclude that is not premature to use either of these models. Therefore, it also makes sense to evaluate how our observations affect the existing model and which differences are induced.

This argument does of course not invalidate the correct critique that neither of the two models have been tested and validated empirically regarding their actual capacity to quantify the connectivity effects on water flow and transport of agrochemicals beyond the few observational studies that triggered this kind of connectivity assessment in the first place. This aspect needs to be clearly communicated.

In summary, we adapted the paper as follows to address the concerns of Reviewer 2 (these changes partially also address the issues raised by the first reviewer):

- The limitations of existing large-scale models are clearly stated and the research need is emphasised.
  → L91-97 (TC: 91-99)
- The objective of evaluating the consequences is clearly stated in the Introduction.
  → L103-104, L111-112 (TC: 105-106, L113-114)
- The focus of the large-scale aspect is on the comparison with the existing model. To be consistent with this objective, we replaced the current Fig. 8 with a map depicting the differences between the two models. The focus is clearly on how the additional empirical observations influence the model predictions.
  → L549-570, Figure 8 (TC: L584-605, Figure 8)
- Along the same lines, we skipped those parts of the text that discuss the findings specifically for the pesticide issues.
  → TC: L614-623 were deleted

**Specific comments**

**(1) Lines 106 ff: Shortcut definition should be moved to section 2.2 Assessment of hydraulic shortcuts:**

We moved this. → L137-150 (TC: L140-152)

**(2) Lines 125 ff: The probability of selection was proportional to the total area of arable land...How was this represented in the random selection?**

We are not sure if we understand the question correctly, but try to answer it as good as possible. We performed a weighted random selection from a list of all catchments. The weights of each catchment equalled the total area of arable land in the catchment. Specifically, we were using the python function "numpy.random.choice":

```
numpy.random.choice(a = catchment_id_list, size = 20, p = catchment_area_list)
```

**(3) Lines 156 ff: How did you prevent selection bias due to drainage plans? I.e. how did you rule out, that no available drainage plan did not correspond to no existing drainage system**

We tried to reduce the impact of selection bias as much as possible by using three different acquisition methods. If drainage plans are not available in a certain catchment, the other two methods are to some degree filling this gap and accordingly reducing the selection bias. As shown in Table 5, the drainage plan mapping method had a lower recall than the aerial image mapping methods. Additionally, also the number of shortcuts identified by aerial

images is much higher than by drainage plans. Therefore, the aerial image mapping method is more important for the overall result and we expect the selection bias due to drainage plans to be small. However, we are sure that we still missed some of the shortcuts and addressed this in L423-424 (TC: 437-439) by writing that the numbers reported are a lower boundary estimate. This is also in the discussion section (L612-627; TC: 660-675).

**(4) Line 243: Elaborate under which circumstances hedge infiltration may be active or inactive**

We could imagine various factors affecting the runoff capturing efficiency of a hedge. For example, the width of the hedge, shrub species, or the degree of runoff concentration. We did choose this parameter to have a binary distribution since no further information on the hedges (such as hedge width) were available. As shown in the sensitivity analysis (Figure S22 and S23), the hedge width parameter only has a minor influence on the overall results. Therefore, we also did not spend time in refining this parameter further.

**(5) Line 271 ff: Conceptually, the parameter "maximal flow distance" should depend on soil properties and cultivation phase, was this considered?**

This was not considered directly, but indirectly. Since no data on soil properties is available on national scale in Switzerland, we tried to identify potential differences in soil properties by calculating the topographic wetness index (TWI) (see L304 ff.; TC: 311 ff.). We did not find any systematic differences between the TWI distributions of directly and indirectly connected areas (see L507 ff.; TC: 531 ff.). We therefore also do not expect systematic differences in soil properties between directly and indirectly connected areas. We did not address the cultivation phase specifically. This influence factor is expected to cause large differences in maximal flow distances in time and space. However, we again do not expect systematic differences of this influence factor between directly and indirectly connected areas.

Consequently, we expect that the maximal flow distances found on directly and indirectly connected areas are not systematically different from each other.

**(6) Line 283: It is unclear how "all possible flow distances were evaluated" with 100 model realizations**

We agree that this is not clearly formulated in the manuscript, and Reviewer #1 also stated this. For each of the 100 model realizations, the maximal flow distance was varied between 2m and the maximal flow distance in the catchment in steps of 2m. In each step, areas with flow distances longer than the maximal flow distance were defined as "not connected". Afterwards, for each of these steps, fractions of "directly connected", "indirectly connected", and "not connected" were calculated.

However, in the manuscript we then aggregated these results into the following categories: <100m, 100 to 200m, 200m to 500m, and >500m. Therefore, the previous formulation was not well chosen.

We adapted the manuscript to make the procedure easier to understand.

→ L275-278, 463-464, Figure S24 (TC: L282-297, L484-485, Figure S24)

**(7) Line 295: Suggested section title change to 'hydrological boundary conditions'**

We think the more specific term "hydrological activity" fits better here, since this term is usually used in the context of critical source areas, for example see: Pionke et al. (2000)

**(8) Line 312 ff.: Although crop type and rain intensity probably don't differ systematically within one catchment, they do impact runoff generation. Should this not be systematically evaluated e.g. across catchments?**

Yes, these factors affect runoff generation and are expected to differ systematically between catchments. As also mentioned in L299-301 (TC: L306-308), this manuscript focuses on comparing indirect surface runoff to direct surface runoff. We therefore were looking for systematic differences between indirectly and directly connected areas. Comparing surface runoff generation across catchments was not within the scope of this manuscript. Currently, except from rainfall data, the data availability for such a comparison is not given. For example, crop data are currently not available in sufficient resolution (see also L349-350, TC: 362-364), soil maps are not available on a national scale. It is also unknown how farming practices (e.g. pesticide application or soil management) differ between catchments.

We however agree that this could be an interesting direction to go and added a paragraph in the "further research" section. → L773-779 (TC: L839-845)

**(9) Line 384 ff / Tab. 3: why is the destination of such a large number of drainage structures unknown? The maps (Figs. 5, S 2.2.1) suggest line / network structure for most inlets, was the outlet of these unclear? How were unknown drainage locations treated in the connectedness classification?**

Three reasons were mainly responsible for this problem:

1) There was no drainage plan available in the whole catchment.
2) Drainage plans were available in the catchment, but did not cover the specific region where the potential shortcut was located.
3) Drainage plans were available in the catchment and did cover the specific region where the potential shortcut was located, but the potential shortcut and its drainage structure were not shown on the plans. (They were however identified during the field survey or on the aerial images.)

For the inlets with known drainage locations 99 % were connected to the surface waters (87 %) or via WWTP/CSO (12 %). Therefore, we assumed in the connectivity model that all shortcuts with unknown drainage locations drain to surface waters (see L257-258; TC: L261-262).

**(10) Lines 419 ff: In Lines 147 ff the field survey was described as "we walked along roads and paths and mapped all the potential shortcut structures." How was mapping accuracy of inlets (5%) and manholes (25.5%) on fields and other areas validated? Lower accuracy esp. in fields with dense vegetation seems likely.**

The accuracy of field mapping could not be validated since we would need a "ground truth" for this, which was not available. We agree that lower accuracy in fields with dense vegetation is likely and discussed this issue in L618-621 (TC: L666-669).

**(10 cont.) How are false positives from mapping ruled out to be false negatives (overlooked structures) from the field survey? How are false negatives from the aerial images quantified altogether?**

From your question, we noticed that Table 5 is not clear enough. As you state correctly, we cannot differentiate between false positives from the aerial image/drainage plan method and false negatives (=overlooked structures) from the field survey when looking at the *identification* of a shortcut structure. For the *identification,* we therefore only report the recall. However, given that a shortcut structure is identified, we can analyse if it is *classified* correctly by the aerial image/drainage plan method (see also L412-416). For the *classification,* we can also report false negatives. For example, a shortcut structure was identified by the aerial image method and was classified as a maintenance manhole. In fact, the field survey showed that the structure showed that the structure is an inlet. This would correspond to a false negative classification.

We adapted Table 5, to make the difference between identification and classification accuracies more clear. → Table 5 (TC: Table 5)

**(11) Line 463: Are the 21 % in Müswangen and 97 % in Boncourt medians / means of the MC ensemble results?**

Yes, those are the medians. We adapted the sentence as follows:

"However, this fraction varies strongly between the study areas, with median fractions ranging from 21 % in Müswangen to 97 % in Boncourt." → L464-465 (TC: 485-486)

**(12) Lines 504 ff: Apart from distribution similarity, how do wetness index and slope affect connected area proportions? I.e. do catchments with higher "hydrological activity" exhibit higher connected proportions?**

Our connectivity model produces something like a "theoretical connectivity map", i.e. the areas reported are connected under the assumption that surface runoff is produced and that this surface runoff is not infiltrating before reaching the recipient area. In contrast, the "effective connectivity" depends on the amount of surface runoff produced and the amount of surface runoff infiltrating before reaching the recipient area.

Wetness index and slope are positively correlated to the probability of an area to be hydrologically active, i.e. more surface runoff is produced. Additionally, they are negatively correlated to the probability that surface runoff infiltrating before reaching the recipient area.

Accordingly, areas with higher wetness index and slope are expected to exhibit a higher "effective connectivity". In fact, those considerations were the reason for performing the analysis described in L503 ff. (TC: L528 ff.)

**(13) Lines 515 ff: I suggest to quantify the deviation between NSCM and LSCM with RSME or another goodness of fit measure.**

We think this is a valuable suggestion. Therefore, we calculated the RSME between the NECM and the LSCM, and between the NSCM and the LSCM. The manuscript was adapted as follows:

"The differences to the LSCM were strongly reduced by this transformation. The root-mean-square error (RSME) reduced from 17 % to 9.5 % for directly connected fractions, from 12 % to 7.6 % for indirectly connected fractions, and from 18 % to 7.6 % for not connected fractions." → L524-527 (TC: L549-553)

**(14) In Fig 7 the mean fraction not connected of LSCM appears to be roughly 15% higher than of NSCM but the text states it is 3% larger (l. 545), why do text and figure differ?**

There is an error in the legend of Figure 7. While the colours were described correctly in the figure description, in the legend (on the top right of the figure) the colours "red" and "blue" were interchanged. We corrected Figure 7 in the manuscript accordingly.

→ Figure 7 (TC: Figure 7)

**(15) Lines 554 ff: I disagree, that "map corresponds to a risk map of pesticide transport via hydraulic shortcuts". Although hydraulic shortcuts contribute to the risk, surface runoff proportion and volume, pesticide application intensity and retention in treatment facilities contribute to this risk as well. This is acknowledged in the discussion of NSCM (Lines 700 ff). (See also general comment above)**

We agree with this point. With revising and shortening the second part of the paper (as written in our answer to the general comments of Reviewer #2), we replaced Figure 8 and removed this sentence (TC: 614-623 removed).

**(16) Line 599: phrase starting with "In Buchs,..." is unclear.**

We rephrased this to:

"In Buchs, around 60 % of the channel drain and ditch length consists of ditches that cannot be clearly distinguished from small streams." → L608-609 (TC: L654-655)

**(17) Line 633 – 687 the parameter discussion is somewhat self-referential without calibration data: Road carving depth, sink depth, and shortcut definition should be evaluated / calibrated by observed events. Assumptions such as 'higher DEM resolution is better' or 'manhole sinks should not be filled completely' seem plausible but can't be substantiated by the results of this research. The impact of hydrological activity parameters may not differ between directly and indirectly connected areas of the same catchment, but among catchments and should be evaluated accordingly.**

We agree that our quantitative results cannot substantiate statements that higher DEM resolution was better. However, the field observations – e.g. based on visual inspections of sediment deposition along roads – clearly revealed that the microtopography can play a major role in controlling the flux of water, solutes and sediments into shortcuts. Furthermore, one has to be aware that these parameters are global parameters in the model despite the fact that there might be regional differences. The optimal road carving depth for example may differ according to topography, regional construction standards, etc. Given this situation, we also think that the optimal way to go would be to calibrate these parameters by observed events. However, on a scale of 20 catchments this would be an extremely laborious task.

Additionally, field observations of these parameters also underlie high uncertainties. For example, sink depths are strongly variable in time and space. We therefore aimed on discussing other options that could be used to improve our results. To clarify that these statements are not findings that can be substantiated by our results, but a discussion of improvement options, we modified the manuscript as follows:

- Higher DEM resolution: We replaced the word "would" in the sentence in L646-647 (TC: L694-695) by the word "could", since we can't tell from our results that this would actually improve the model.
- Sink filling: Similarily, we replaced the word "can" by the word "might" in L659-660 (TC: 707-708).

**(18) Line 733: Suggested extension: End of pipe measures at shortcut / pipe outlets(treatment, sedimentation, filtration)**

We agree on this and adapted this sentence to:

"Other measures could aim on the shortcut structures themselves (e.g. construction of shortcuts as small infiltration basins, removal of shortcuts, or treatment of water in shortcuts) or on the pipe outlets (e.g. drainage of shortcuts to infiltration basins, treatment of water at the pipe outlet)." → L752-755 (TC: 818-821)

**(19) In my view, the 2nd research question (line 104) can't be fully answered with the present approach. It should be rephrased and / or referred in the conclusions section.**

In our view, we can actually answer the second research question with our approach. We did not find any evidence on systematic differences in hydrological activity. In addition, we do not expect systematic differences in farming practices, precipitation, or crop types between directly and indirectly connected areas. Given the current knowledge, we therefore expect that the proportions of direct and indirect surface runoff related pesticide transport are proportional to the directly and indirectly connected area. However, we think that this is not formulated clear enough and therefore revised the results and conclusions accordingly.

→ Results: L512-515 (TC: 536-539)

→ Conclusions: L805-808 (TC: 872-875)

**(20) Line 1116: the table title is confusing. I assume directly connected local surface connectivity model fraction was correlated to directly connected national erosion connectivity model fraction and so on...**

We agree and adapted the table accordingly. → Table S8 (TC: Table S8)

**References**

Doppler, T., Camenzuli, L., Hirzel, G., Krauss, M., Lück, A., and Stamm, C.: Spatial variability of herbicide mobilisation and transport at catchment scale: insights from a field experiment, Hydrol Earth Syst Sc, 16, 1947-1967, https://doi.org/10.5194/hess-16-1947-2012, 2012.

Frey, M. P., Schneider, M. K., Dietzel, A., Reichert, P., and Stamm, C.: Predicting critical source areas for diffuse herbicide losses to surface waters: Role of connectivity and boundary conditions, J Hydrol, 365, 23-36, 2009.

Leu, C., Singer, H., Stamm, C., Müller, S. R., and Schwarzenbach, R. P.: Variability of Herbicide Losses from 13 Fields to Surface Water within a Small Catchment after a Controlled Herbicide Application, Environ Sci Technol, 38, 3835-3841, https://doi.org/10.1021/es0499593, 2004.

Mutzner, L., Bohren, C., Mangold, S., Bloem, S., and Ort, C.: Spatial Differences among Micropollutants in Sewer Overflows: A Multisite Analysis Using Passive Samplers, Environ Sci Technol, 54, 6584-6593, 10.1021/acs.est.9b05148, 2020.

Pionke, H. B., Gburek, W. J., and Sharpley, A. N.: Critical source area controls on water quality in an agricultural watershed located in the Chesapeake Basin, Ecol Eng, 14, 325-335, Doi 10.1016/S0925-8574(99)00059-2, 2000.

---

## Author Response (AR2)

**Author's response**

We first would like to thank the Editor, and both Referees for reading our manuscript again, and for providing additional comments. In our opinion, this helped again to further improve the manuscript.

In the following, our point-by-point response to the reviews is given. We first answer on the comments of the Editor, followed by the answer to the comments of Referee #1 and Referee #2. The line numbers refer to the track-changes document.

**Answer to Editor**

In the first paragraph, the Editor emphasizes one of the main comments of Referee #1. This comment states that the second research question is not correctly formulated and raises wrong expectations. We agree on this point and modified the manuscript accordingly. (TC: L110-112). We also addressed all other comments of Referee #1 (see answer to Referee #1).

In the second and third paragraph, the Editor addresses the concerns of Referee #2 about the national extrapolation. The Editor states that she thinks this part should not be taken out of the manuscript, but that the concerns of Referee #2 about the uncertainty of the national extrapolation and the issues about lack of validation should be addressed more in the manuscript. We agree that the lack of validation is an important aspect for discussion. We addressed this aspect in more detail in the discussion sections "Extrapolation to the national level" (TC: L726-744), and "Implications for practice" (TC: L767-775). Additionally, we added this aspect to the conclusion section (TC: L886-887) and included a respective sentence in the abstract (TC: L35-36).

In the third paragraph, the Editor additionally states that it should be outlined more clearly how more reliability on the national assessment could be achieved. We addressed this point in the manuscript by extending the paragraph on model validation in the "Further research" section of the discussion (TC: L787-842).

**Answer to Anonymous Referee #1**

The main concerns of Referee #1 relate to the extrapolation to the national level. The referee mainly criticises that the added value of this part is small, and that the uncertainty of this approach is large. Additionally, the referee states that one should be careful with making recommendations to authorities based on a model which is not validated with field measurements.

We still think that this part has additional scientific value, since it updates a previous national scale model based on additional empirical data. We however agree on the concerns that this extrapolation is subject to substantial uncertainties and that one should be careful about the content and phrasing of recommendations to authorities. We tried to give these aspects more weight in the manuscript and extended the paragraphs on model validation (see answer to Editor). Particularly, we stated the lack of validation more clearly in the "implications for practice" section.

**Answer to Anonymous Referee #2**

In the following we will give a point by point answer to the comment of Referee #2.

**In my previous comment 19, I argued that the 2nd research question could not be fully answered. […] I suggest to adapt question 2) as follows: What is the contribution of hydraulic shortcuts to surface runoff connectivity and what are implications for surface-runoff related pesticide transport?**

We agree on this argument and adapted the manuscript accordingly (TC: L110-112).

**And to extend L 536 – 539 ACT as follows: Analogously, if other boundary conditions of pesticide transport remain unchanged, directly and indirectly transported pesticide loads are expected to be proportional to directly and indirectly connected crop areas.**

We agree on this comment and adapted the manuscript accordingly (TC: L521-525).

**As a general remark: If available, a revision of phrasing, esp. of the M&M section, could make the text easier to read. Some needlessly complicated sentence constructions are: L 212 – 214 […], L 236 – 239 […], L 328 – 330 […], L 362 – 365 […]**

We agree that some of the sentences in the Methods section were too complicated and rephrased the above-mentioned sentences (TC: L212-215, L236-240, L325-329, L356-359). We also rephrased some additional sentences for better readability.